# Free energies of membrane stalk formation from a lipidomics perspective

Chetan S. Poojari[1], Katharina C. Scherer [1] & Jochen S. Hub [1✉]

Many biological membranes are asymmetric and exhibit complex lipid composition, comprising hundreds of distinct chemical species. Identifying the biological function and advantage of this complexity is a central goal of membrane biology. Here, we study how membrane complexity controls the energetics of the first steps of membrane fusions, that is, the formation of a stalk. We first present a computationally efficient method for simulating thermodynamically reversible pathways of stalk formation at coarse-grained resolution. The method reveals that the inner leaflet of a typical plasma membrane is far more fusogenic than the outer leaflet, which is likely an adaptation to evolutionary pressure. To rationalize these findings by the distinct lipid compositions, we computed ~200 free energies of stalk formation in membranes with different lipid head groups, tail lengths, tail unsaturations, and sterol content. In summary, the simulations reveal a drastic influence of the lipid composition on stalk formation and a comprehensive fusogenicity map of many biologically relevant lipid classes.

---

[1] Saarland University, Theoretical Physics and Center for Biophysics, Saarland University, Saarbrücken, Germany. ✉email: jochen.hub@uni-saarland.de

Eukaryotic cellular membranes contain more than ten lipid classes, while each class comprises hundreds of different chemical species[1,2]. The complexity of membranes is further increased by the membrane asymmetry, that is, by distinct lipid compositions in the two leaflets. In the plasma membrane of mammals, sphingolipids are typically enriched in the outer leaflet of the plasma membrane, whereas phosphatidylethanolamine (PE) and phosphatidylserine (PS) are enriched in the inner leaflet[3]. Apart from different head groups, lipid species differ by the length and unsaturation of the fatty acid tails. Recent lipidomics studies showed that the degree of polyunsaturation in the inner leaflet is approximately twice that of the outer leaflet[4]. Understanding why biological cells synthesize and maintain this complex lipid repertoire, that is, defining the biological function and advantage of specific lipid compositions, remains a central goal of membrane biophysics.

How complex lipid compositions control the early stages of membrane fusion has not been systematically addressed. Fusion is critical for many processes that involve the transport of cargos across membranes such as exocytosis, neurotransmission, infection of cells by enveloped viruses, fertilization, and intracellular transport[5,6]. Fusion involves high kinetic barriers because it requires to overcome the hydration repulsion between the membranes and to form intermediates with highly curved membranes. Cells and viruses use fusion proteins to overcome these kinetic barriers[7,8]. The process starts with two membranes in the lamellar phase at the equilibrium distance of 2-3 nm, followed by bridging of two opposing membranes by lipid acyl chains to establish a point-like protrusion. The lipids of the two contacting, proximal leaflets mix to form the initial transient hemifusion stalk. The stalk expands to allow the formation and expansion of a fusion pore[9–11].

Because intermediate structures along the fusion pathway involve highly curved membranes, the intrinsic curvature of lipids influences the kinetics of fusion. For instance, when added to the proximal leaflets, inverted cone-shaped lysophosphatidylcholine (LPC) inhibits stalk formation because it is incompatible with the large negative curvature at the stalk rim. In contrast, cone-shaped unsaturated PE lipids or diacylglycerol promote stalk formation[10,12,13]. Here, the unsaturated fatty acids render the lipids more cone-shaped because the double bonds increase the tail disorder and, hence, the effective volume of the lipid tails[13]. In addition to such effects by the geometric lipid shape, the increased abundance of double bonds increases the conformational flexibility of the lipids, thereby allowing the membrane to adapt more easily to curvature[14], which may further promote fusion.

Theoretical and computational approaches have established possible fusion pathways and the underlying free energy landscapes, however, only few studies focused on the role of the lipid composition during fusion. Early studies employed continuum descriptions, which model the role of lipids in terms of the effective spontaneous curvature and membrane rigidity[15], as well as minimal coarse-grained lipid models in conjunction with Monte-Carlo, field-theoretic, or Brownian dynamics methods[16–18]. Complementary, to account for the chemical specificity of lipid–lipid interactions, fusion was studied with atomistic and coarse-grained (CG) molecular dynamics (MD) simulations[19–21], discussed in several excellent reviews[22–24]. PE lipids were found to enhance fusion during simulations as expected from their negative intrinsic curvature[25–27], however the effects of other head groups, sterols, or degrees of unsaturation have hardly been considered despite their abundance in biological membranes.

Computationally efficient free energy calculations of fusion require the definition of one or several reaction coordinates (or order parameters) along the fusion pathway; however, finding good coordinates for complex transitions is non-trivial. With poor coordinates, hysteresis problems emerge, barriers may be integrated out, and simulations proceed along non-reversible pathways. To avoid such problems, a recent elegant MD study used the string method to optimize the minimum free energy pathway of stalk formation, parametrized by the three-dimensional (3D) density of the apolar lipid tail beads[27]. However, because the string method is computationally demanding, it becomes prohibitive for high-throughput studies. In this study, we introduce a reaction coordinate for stalk formation that allows computationally highly efficient and thermodynamically reversible free energy calculations of stalk formation using common umbrella sampling simulations. The potentials of mean force (PMFs) along the coordinate, computed with Martini coarse-grained models, show that the inner leaflet of a common plasma membrane is far more fusogenic than the outer leaflet. To rationalize these finding, we screen the fusogenicity of lipids by varying the hydration level between the two membranes, the lipid headgroup size and charge, acyl chain length, and unsaturation levels. In addition to phosphatidylcholine (PC), PE, PS, and phosphatidylglycerol (PG) lipids, we also considered cholesterol, lysolipids, fatty acids in protonated and deprotonated forms, phosphatic acid, ceramide, diacylglycerol, and sphingomyelin. The calculations provide a comprehensive view on the role of lipid complexity during the first steps of membrane fusion.

## Results

**Highly efficient free energy calculations of stalk formation.** To avoid hysteresis problems during PMF calculations, it is mandatory to define a good reaction coordinate. We propose using a reaction coordinate for stalk formation that was originally introduced to study the formation of aqueous pores over membranes[28,29]. The coordinate, named "chain coordinate" $\xi_{ch}$, quantifies the degree of connectivity between two compartments. In brief, $\xi_{ch}$ is defined with the help of a cylinder that spans the head group and water regions of the proximal compartment (Fig. 1A). The cylinder is decomposed into $N_s$ slices, and $\xi_{ch}$ is defined as the fraction of slices that are filled by apolar lipid tail beads. By pulling the system along the coordinate, the slices are filled one-by-one, thereby forming a continuous apolar connection between the two fusing membranes, as required for stalk formation (Fig. 1A–C). The use of a cylinder ensures that two opposing hydrophobic protrusions from the two leaflets into the proximal water compartment are localized in the lateral plane, but the cylinder does not control the shape or the radius of the stalk. More details are provided in the Supplementary Information Methods.

We used umbrella sampling along $\xi_{ch}$ to compute the PMF of stalk formation in conjunction with the coarse-grained Martini lipid force field. Figure 1D presents PMFs of stalk formation across membranes of POPC lipids using different degrees of hydration in the proximal water compartment, spanning 1.25 to 7 water beads per lipid, corresponding to 5 to 28 water molecules per lipid according to the 4:1 mapping of Martini. Here, the length of the $\xi_{ch}$-defining cylinder was chosen such that $\xi_{ch} \approx 0.2$ corresponds to flat unperturbed membranes, implying that 20% of the cylinder slices are filled by lipid tail beads. $\xi_{ch} \approx 1$ corresponds to a fully formed stalk. The PMFs demonstrate that the free energy of the stalk $\Delta G_{stalk}$ relative to the flat membranes greatly depends on the degree of hydration, spanning values from −30 kJ/mol up to 180 kJ/mol (Fig. 1D, inset). With fewer than 2 water beads per lipid, the stalk is separated from the flat membrane by a barrier or transition state, indicating that the stalk is metastable (long-living); with less than 1.5 water beads per lipid, the stalk even forms the free energy minimum. Such negative $\Delta G_{stalk}$ values at very low hydration are rationalized by the energy stored in

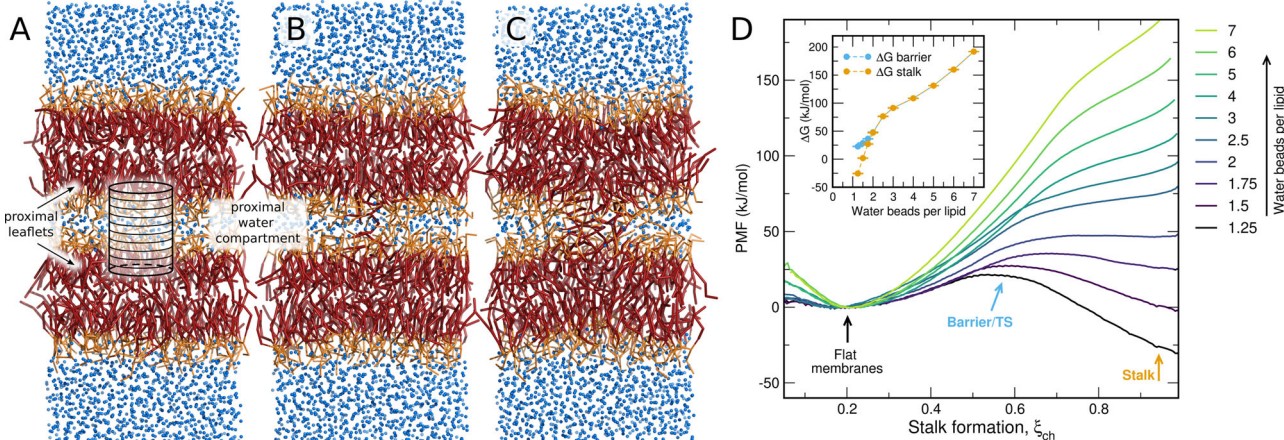

**Fig. 1 Efficient potential of mean force calculations of stalk formation. A–C** Martini simulation system with membranes of pure POPC and 1.5 water beads per lipid in the proximal water compartment, showing representative frames of **A** the flat membrane **B** the transition state (TS), and **C** the stalk. The reaction coordinate $\xi_{ch}$ for stalk formation quantifies the degree of connectivity between the two hydrophobic membrane cores, defined with the help of a sliced cylinder as illustrated in **A**. **D** PMFs of stalk formation for membranes of pure POPC, with increasing hydration in the proximal compartment of 1.25 to 7 water beads per lipid, corresponding to 4 to 21.2 water beads per nm$^2$ (see color code). Inset: Free energy of the stalk $\Delta G_{stalk}$ and of the stalk nucleation barrier ($\Delta G_{barrier}$, if present) versus water beads per lipid in the proximal compartment, as taken from the PMFs. Error bars computed by bootstrapping denote 1 standard error (SE). Source data are provided as a Source Data file.

the hydration repulsion between the two membranes. Upon stalk formation, the hydration repulsion energy is partly relieved, thereby decreasing the free energy of the stalk relative to the flat membranes. The marked dependence of $\Delta G_{stalk}$ on the degree of hydration agrees with earlier models based on elasticity theory[30] and with a recent simulation study, which used the string method together with the 3D lipid tail density as order parameter to derive the minimum free energy path of stalk formation[27]. However, with the identical simulation system, the free energies for the stalk suggested by our PMFs are significantly lower (Supplementary Fig. 2), possibly because our stalk state given by $\xi_{ch} \approx 1$ includes more conformational freedom than a stalk definition via a specific 3D density used in Ref. [27].

The Martini model allows semi-quantitative simulations, suggesting that Martini yields trends often correctly[31]; however, the exact free energy values must often be taken with care and may depend on the Martini version. To test how the stalk free energies depend on the Martini version, we re-computed the PMFs of stalk formation for POPC with the beta-3.2 release of Martini 3.0 (3.0beta) instead of Martini 2.2 (Supplementary Fig. 3). The shape of the PMFs and the dependence on hydration are well preserved among different Martini versions. However, the free energy of the stalk with Martini 3.0beta is reduced by ~ 30 kJ mol$^{-1}$ relative to Martini 2.2, suggesting that the newer Martini version is more fusogenic. Simulations with four additional lipid types confirm this trend (Supplementary Fig. 4). This finding suggests that the free energies reported in this study should be interpreted in terms of trends (with hydration, degree of unsaturation etc.) and not in terms of precise free energy values.

**Kinetics of stalk formation agree with the PMFs.** As a critical test for the quality of the reaction coordinate and, thereby, for the validity of the PMFs, we carried out free simulations of stalk formation and stalk closure (Fig. 2). We simulated a system for which $\Delta G_{stalk} \approx 0$ kJ/mol according to the PMF (Fig. 2B). Among a total simulation time of 800 $\mu$s, we observed 8 transitions of stalk opening and 7 transitions of stalk closure, corresponding to rates of $k_{stalk} = 16$ ms$^{-1}$ and $k_{closure} = 23$ ms$^{-1}$, respectively. Hence, the free simulations suggest a free energy of stalk formation of $\Delta G_{stalk} = -k_B T \ln(k_{stalk}/k_{closure}) = 0.9$ kJ/mol. The

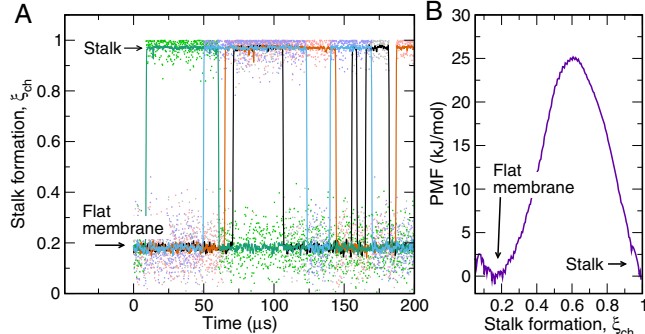

**Fig. 2 Unbiased simulations confirm the free energies of stalk formation. A** Free simulations of stalk formation and closure. Colors indicate four independent simulations of 200 $\mu$s each. $\xi_{ch} \approx 0.2$ and $\xi_{ch} \approx 1$ correspond to the flat membrane and the stalk, respectively. Eight transitions of stalk formation and 7 transitions of stalk closure occurred within a total simulation time of 800 $\mu$s. The simulation system contained purely POPC lipids and 1.8 water beads per lipid between the proximal leaflets (5.6 water beads per nm$^2$). Lipids were modeled with the beta-3.2 release of Martini 3.0. **B** PMF of stalk formation for the same system, revealing a free energy of stalk formation of $\Delta G_{stalk} \approx 0$ kJ/mol, in excellent agreement with the free simulations. Source data are provided as a Source Data file.

excellent agreement with the PMF suggests that the PMF is not affected by hysteresis problems but reports the correct $\Delta G_{stalk}$ as given by the Martini force field. Using transition state theory, the rates are expected to follow $k = \nu\, e^{-\Delta G^{\ddagger}/k_B T}$, where $\nu$ is the attempt frequency, $\Delta G^{\ddagger}$ the free energy barrier in Fig. 2B, and $k_B$ and $T$ are the Boltzmann constant and temperature, respectively. The rates from the free simulations together with the barrier from the PMF imply $\nu \approx 0.3$ ns$^{-1}$, i.e., one attempt per 3 ns. Interestingly, this time scale resembles the time scales required for (i) head groups to rearrange along the membrane normal and (ii) for lipids to travel a typical lipid–lipid distance (Supplementary Fig. 19 and Supplementary Discussion). Hence, the attempts for stalk formation in the context of transition state theory may be interpreted as rearrangements owing to conformational sampling of lipids, which occurs on the nanosecond timescale.

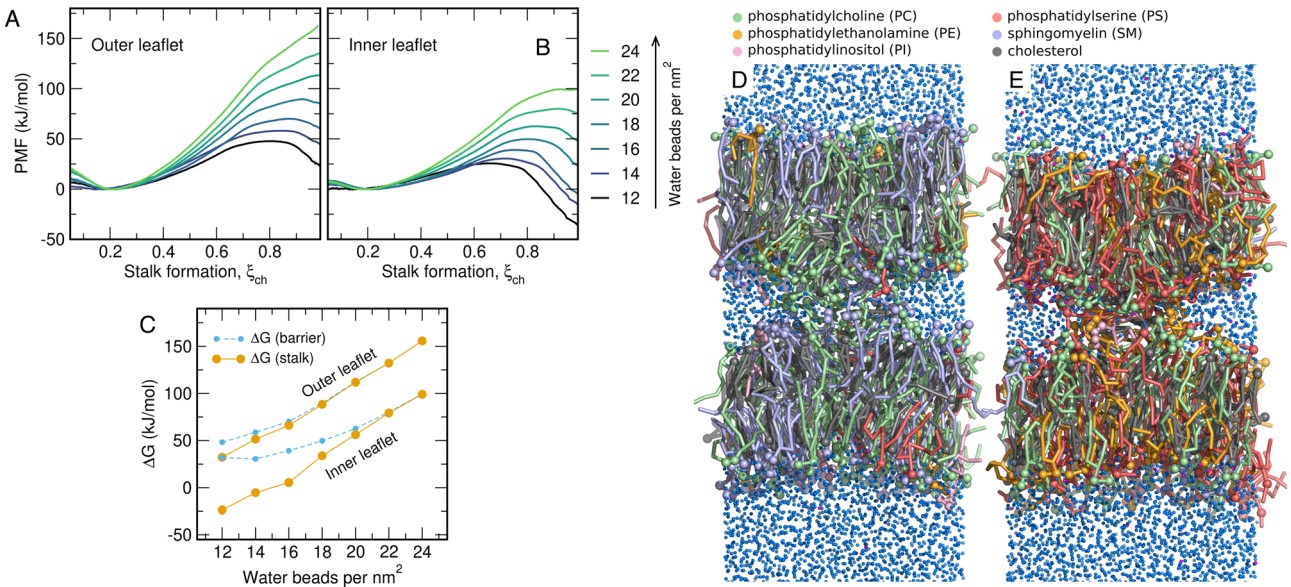

**Fig. 3 Stalk formation between membranes with complex lipid composition. A**, **B** PMFs of stalk formation with lipid composition taken from the plasma membrane **A** outer leaflet or **B** inner leaflet. $\xi_{ch} \approx 0.2$ and $\xi_{ch} \approx 1$ correspond to the states with planar membranes and with the stalk, respectively. PMFs were computed with increasing hydration between the proximal leaflets between 12 and 24 water beads per nm². **C** Free energy of the stalk (orange) and of the barrier for stalk formation (blue), taken from the PMFs in A/B. Error bars computed by bootstrapping denote 1 SE. **D** Simulation frame of the open stalk in simulation with outer leaflet composition and **E** inner leaflet composition. Lipids are shown as colored sticks, headgroup beads as spheres (for color code, see legend). Water and sodium beads are shown as small blue and magenta spheres, respectively. Source data are provided as a Source Data file.

**The inner leaflet of the plasma membrane is far more fusogenic than the outer leaflet**. Many critical fusion events occur with the plasma membrane. During exocytosis, transport vesicles fuse with the plasma membrane, where they first form a stalk with the inner plasma membrane leaflet. During viral infection, the envelope of certain viruses fuses with the plasma membrane, thereby forming a stalk with the outer leaflet. Like many biological membranes, the plasma membrane is asymmetric, i.e., the two leaflets exhibit different lipid compositions. Further, the plasma membrane reveals a complex lipid composition, including various head group types, tail lengths, degrees of tail unsaturation, and steroid content [4].

To reveal how the lipid composition of a complex biological membrane determines the free energies of stalk formation, we set up systems with symmetric membranes, but with the lipid composition mimicking either the outer or inner leaflet of a mammalian plasma membrane[4]. The membranes contained phosphatidyl-choline (PC), -ethanolamine (PE), -inositol (PI), -serine (PS), sphingomyelin (SM), and cholesterol, as well a various tail lengths and degrees of unsaturation, taken from a recent lipidomics study[4] (Fig. 3D/E, Table 1 and Supplementary Table 1). The PMFs of stalk formation were again computed for various degrees of hydration in the proximal compartment quantified by the number of water beads per membrane area (Fig. 3A/B and Supplementary Fig. 5). Remarkably, the free energy of stalk formation between membranes with the outer leaflet composition is larger by ~ 50 kJ/mol as compared to the membrane with the inner leaflet composition, irrespective of the degree of hydration (Fig. 3C, orange dots). The same trend is observed for the free energy barrier for stalk formation (if present, Fig. 3C, blue dots). Hence, the inner leaflet is far more fusogenic than the outer leaflet of the plasma membrane. These different fusogenicities may reflect adaptation to the evolutionary pressure: Efficient fusion with the inner leaflet is required for exocytosis, in particular for rapid fusion of synaptic vesicles with the plasma membrane of the synapse. In contrast, an increased resistance against fusion with the outer leaflet may protect the cell against viral infection.

**Table 1 Summary of lipid composition and properties in models for the outer and inner leaflet.**

|  | Outer | Inner |
|---|---|---|
| PC | 25% | 15% |
| PE | 2% | 13% |
| SM | 25% | 2% |
| PI | 2% | 2% |
| PS | 3% | 25% |
| Cholesterol | 43% | 43% |
| Saturated beads | 89% | 77% |
| Unsaturated beads | 11% | 23% |
| Av. tail length (beads) | 3.9 | 4.2 |

**Free energies of stalk formation are strongly influenced by tail unsaturation, tail length, and headgroup type**. Why is the inner plasma membrane leaflet more fusogenic than the outer leaflet? As listed in Table 1 and Supplementary Table 1, the inner leaflet model contains more PE and PS lipids, whereas the outer leaflet contains more PC and SM lipids. In addition, the lipid tails of the inner leaflet are more unsaturated as compared to the outer leaflet, as given by the fraction of beads modeling unsaturated tails (23% versus 11%), and the lipid tails of the inner leaflet are longer on average (4.2 versus 3.9 beads). Owing to this complexity, it is difficult to extract the key lipid properties underlying the distinct fusogenicities.

To disentangle the influence of tail unsaturation, tail length, and type of head group we computed 110 additional PMFs of stalk formation. These calculations were accessible only thanks to the computational efficiency of the PMF calculations. We systematically varied either (i) the degree of unsaturation by simulating fully saturated up to poly-unsaturated tails, (ii) the tail length from 3 to 6 coarse-grained beads, thereby modeling approximately 14 to 26 carbon atoms per tail, or (iii) the type of head group between PS, PG, PC, and PE (Fig. 4A–C). The PMFs

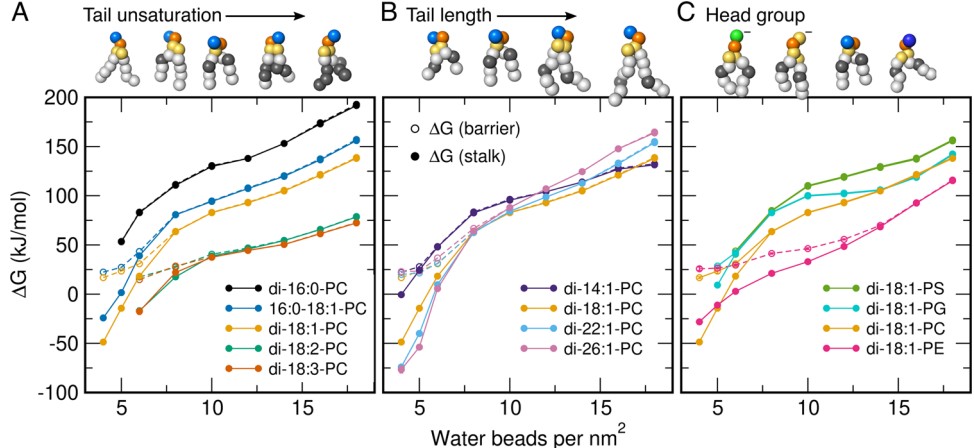

**Fig. 4 Impact of tail unsaturation, tail length, and head group.** Free energies of stalk formation $\Delta G_{stalk}$ (solid circles) for single-lipid membranes composed of various lipid types, plotted as a function of increasing hydration between the proximal leaflets. Legends indicate the approximate atomistic correspondence of the Martini models. Open circles show the free energy barrier for stalk formation (if present). Beads of Martini lipid models are colored as follows: hydrophobic saturated (white), hydrophobic unsaturated (gray), glycerol (yellow), phosphate (orange), choline (light blue), serine (green), ethanolamine (dark blue). **A** Lipids with increasing tail unsaturation, **B** with increasing tail length, and **C** with different head groups. Source data are provided as a Source Data file.

were again computed for various degrees of hydration between 4 and 18 water beads per nm$^2$ (Supplementary Figs. 6 and 7). Molecular representations of the stalks are shown in Supplementary Figs. 8, 9 and 10.

Apart from the degree of hydration, also the lipid type strongly influences the free energy of the stalk $\Delta G_{stalk}$ and the free energy barrier for stalk formation. Specifically, saturated tails as found in DPPC (di-16:0-PC) disfavor stalk formation, whereas polyunsaturated tails in lipids such as di-18:2-PC greatly favor stalk formation (Fig. 4A). In addition, increasing the tail length may stabilize the stalk at low hydration of ≤6 water beads per nm$^2$ for which the stalk is energetically accessible (Fig. 4B). At increased hydration, however, for which the stalk is unstable, longer tails further destabilize the stalk. Likewise, the type of head group may influence $\Delta G_{stalk}$: the anionic head groups of dioleoyl-PS (DOPS) and DOPG disfavor stalk formation by ~ 25 kJ/mol relative to the zwitterionic head group of DOPC (Fig. 4C). These findings cannot be explained merely by the geometric shape of the lipids because PS, PG, and PC headgroups have a similar size in the Martini model, as evident from very similar areas per lipid (Supplementary Tab. 2). Hence, the electrostatic repulsion between the anionic lipids plays a distinct role in destabilizing the stalk, possibly by effectively inducing a more positive spontaneous membrane curvature. The small zwitterionic PE headgroup facilitates stalk formation relative to the larger PC headgroup by up to 50 kJ/mol, as one may expect from the cone shape of PE lipids that may stabilize the strong negative curvature in the stalk[32]. This trend is only inverted at very low hydration of ≤5 water beads per nm$^2$, where stalk formation between PC membranes is more favorable than between PE membranes (Fig. 4C, yellow and magenta dots). Simulations with PE and PC membranes with alternative tails confirmed that, at most degrees of hydration, PE headgroups typically facilitate stalk formation relative to PC (Supplementary Fig. 6).

Taken together, these data demonstrate that the free energy cost for stalk formation is controlled by a range of lipid properties. Namely, stalk formation is facilitated by (i) increased tail unsaturation, (ii) by longer tails at low hydration between the fusing membranes, (iii) by zwitterionic relative to anionic lipids, and (iv) by the small PE instead of the larger PC headgroup (at most degrees of hydration).

**$\Delta G_{stalk}$ scales linearly with lipid concentration in lipid mixtures.** Next, we investigated how the lipid type and concentration influence stalk formation in lipid mixtures. To this end, we set up addition 76 simulation systems containing POPC as reference lipid plus one type of lipid with increasing mole fraction. A constant degree of hydration with 6 water beads per nm$^2$ was used. In addition to the PC, PG, PS, and PE lipids considered above, and to obtain a comprehensive view on the influence of lipids on stalk free energies, we considered a wide range of additional lipids: cholesterol, lyso lipids, fatty acids, phosphatidic acid (PA), ceramide, diacylglycerol, and sphingomyelin (SM).

Figure 5 presents $\Delta G_{stalk}$ as function of mole fraction $x_{lipid}$ for various lipid types, where $x_{lipid} = 0\%$ corresponds to pure POPC. Typical PMFs are presented in Supplementary Fig. 11. Evidently, mixing a second type of lipid into a POPC membrane may greatly stabilize or destabilize the stalk, while $\Delta G_{stalk}$ depends nearly linearly on the mole fraction of the second lipid. Here, the slope of the $\Delta G_{stalk}$-$x_{lipid}$ curves quantify the sensitivity of stalk formation upon the addition of a second lipid.

As expected[10,33], the geometric shape of the lipid, that is whether the lipid is cone- or inverted cone-shaped, affects $\Delta G_{stalk}$. Only small amounts of ceramide and diacylglycerol greatly stabilize stalks, compatible with their highly negative intrinsic curvature (Fig. 5A, dark blue, dark orange). On the contrary, single-chained lysoPC greatly destabilizes the stalk, compatible with its large positive curvature (Fig. 5A, orange). Single-chained fatty acids stabilize the stalk in the protonated form 16:0-COOH, however they destabilize the stalk in the deprotonated form 16:0-COO$^-$ (Fig. 5A, light blue, purple). Owing to the strongly increased p$K_a$ of fatty acids in lipid membranes relative to bulk water, the protonated, stalk-stabilizing form is predominantly present in biological membranes[34].

In agreement with the findings for single-component membranes presented above, $\Delta G_{stalk}$ is modulated by replacing PC head groups of POPC with other head groups while maintaining the palmitoyl-oleoyl tails. Whereas replacing PC with PA has only a small effect on $\Delta G_{stalk}$, replacing PC with PE favors stalk formation (Fig. 5A, green, Fig. 5B, orange), in agreement with experiments and simulations[26,32]. In contrast, replacing PC with the anionic PS or PG head groups disfavors stalk formation (Fig. 5B, light and dark blue). Likewise, sphingomyelin disfavors

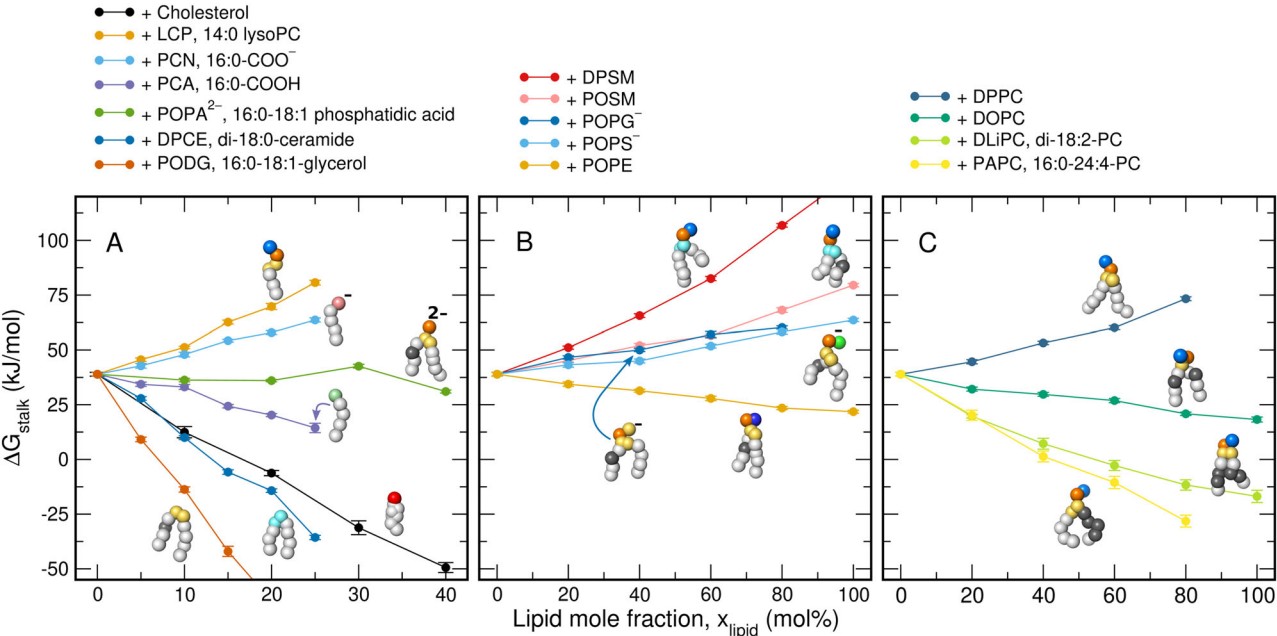

**Fig. 5 Stalk free energies for binary lipid mixtures.** Free energies of stalk formation $\Delta G_{stalk}$ for binary lipid mixtures of POPC plus one additional type of lipid. 0 mol% correspond to pure POPC. The additional lipids are **A** cholesterol, lysoPC, fatty acids, phosphatidic acid, ceramide, diacylglycerol, **B** sphingolipids, phosphatidylglycerol, -serine, and -ethanolamine, **C** phosphatidylcholine with increasing tail unsaturation. For color code, see legend. Martini beads are colored with the scheme used for Fig. 4 and as follows: hydroxyl (red), carboxylate (pale red), carboxyl (pale green), sphingosine (cyan). Error bars computed by bootstrapping denote 1 SE. Source data are provided as a Source Data file.

stalk formation, in particular the fully saturated DPSM. These findings are rationalized by the tendency of sphingomyelin of increasing the order of the aliphatic tails, thereby disfavoring the formation of a defect such as a stalk. In addition, the $\Delta G_{stalk}$ values from lipid mixtures confirm that the addition of (poly-) unsaturated lipid tails facilitate stalk formation, whereas the addition of saturated tails hinder stalk formation (Fig. 5C). Finally, the addition of cholesterol strongly stabilizes the stalk, by up to 45 kJ/mol at 40 mol% cholesterol, in qualitative agreement with results from X-ray diffraction[35].

**Stalk stabilization correlates with specific lipid enrichment in the stalk, except for cholesterol.** To shed light on the molecular mechanism by which different lipids influence $\Delta G_{stalk}$, we computed the lipid densities in simulations with a fully formed stalk (Fig. 6A, B and Supplementary Figs. 13–16). The overall shape of the stalk is largely independent of the lipid composition, except that energetically unstable stalks are thinner as compared to stalks that form the free energy minimum. The shapes reasonably agree with previous simulation studies[18], but the stalks obtained here –even if metastable– are slightly thinner as compared to stalks modeled by continuum descriptions[15], possibly because the lipids were highly disordered inside the stalk in our simulations.

The lipid composition strongly influences the spatial distribution of the individual lipid types between the stalk and the lamellar region. For instance, the stalk-stabilizing DLiPC is enriched at the stalk, whereas stalk-destabilizing DPSM is depleted at the stalk (Fig 6A/B, left column). To test whether these trends hold for other lipids, we quantified the enrichment of a lipid in the stalk as $\rho_{stalk}/\rho_{mem} - 1$, where $\rho_{stalk}$ and $\rho_{mem}$ denote the average lipid mass densities at the stalk and in a region of the flat membrane, respectively (Fig 6A, white and yellow boxes). Hence, a positive and negative values indicate lipid enrichment or depletion at the stalk, respectively. In addition, we quantified the

influence of specific lipids on stalk formation as the slope of a linear least-square fit to the data in Fig. 5, denoted $\Delta\Delta G_{stalk}/x_{lipid}$.

Figure 6 C/D reveals that, among nearly all lipids, lipid enrichment in the stalk anti-correlates with the $\Delta\Delta G_{stalk}/x_{lipid}$. Hence, lipids that either favor negative curvature, increased tail disorder, or both, may accumulate in the stalk, thereby stabilizing the stalk structure. However, there are notable outliers: cholesterol strongly stabilizes the stalk while being only marginally enriched in the stalk[36], possibly because the planar cholesterol molecule disfavors the increased tail disorder in the stalk. These findings suggest that cholesterol does not favor stalk formation owing to an intrinsic negative curvature of cholesterol[37]. To test whether cholesterol takes effect by modulating the hydration repulsion between the membranes, we carried out PMF calculations of dehydrating the proximal leaflet (Supplementary Fig. 12). In qualitative agreement with experimental data[35,38], the PMFs reveal that the addition of cholesterol may reduce the hydration repulsion free energy by tens of kilojoule per mole. Although it is difficult to isolate the role of hydration repulsion on the step of stalk formation simulated here, it is plausible that such reduced hydration repulsion is a key mechanism by which cholesterol influences the overall fusion process (see Discussion).

Another outlier is diacylglycerol PODG, by far the most stalk-stabilizing lipid considered in this study. The lipid densities show that PODG accumulated not only inside the stalk, but mostly on top and below the stalk within the membrane core (Supplementary Fig. 14). The accumulation within the membrane core is possible because the headgroup of diacyl-glycerol is only weakly polar. These findings are in line with experiments, which showed that even small concentrations of diacylglycerol promote the formation of stalk-like structures [39,40]. We hypothesize that a similar mechanism applies during biogenesis of lipid droplets, which are released from the ER membrane via a stalk-like intermediate[41]. Lipid droplets contain large amounts of only

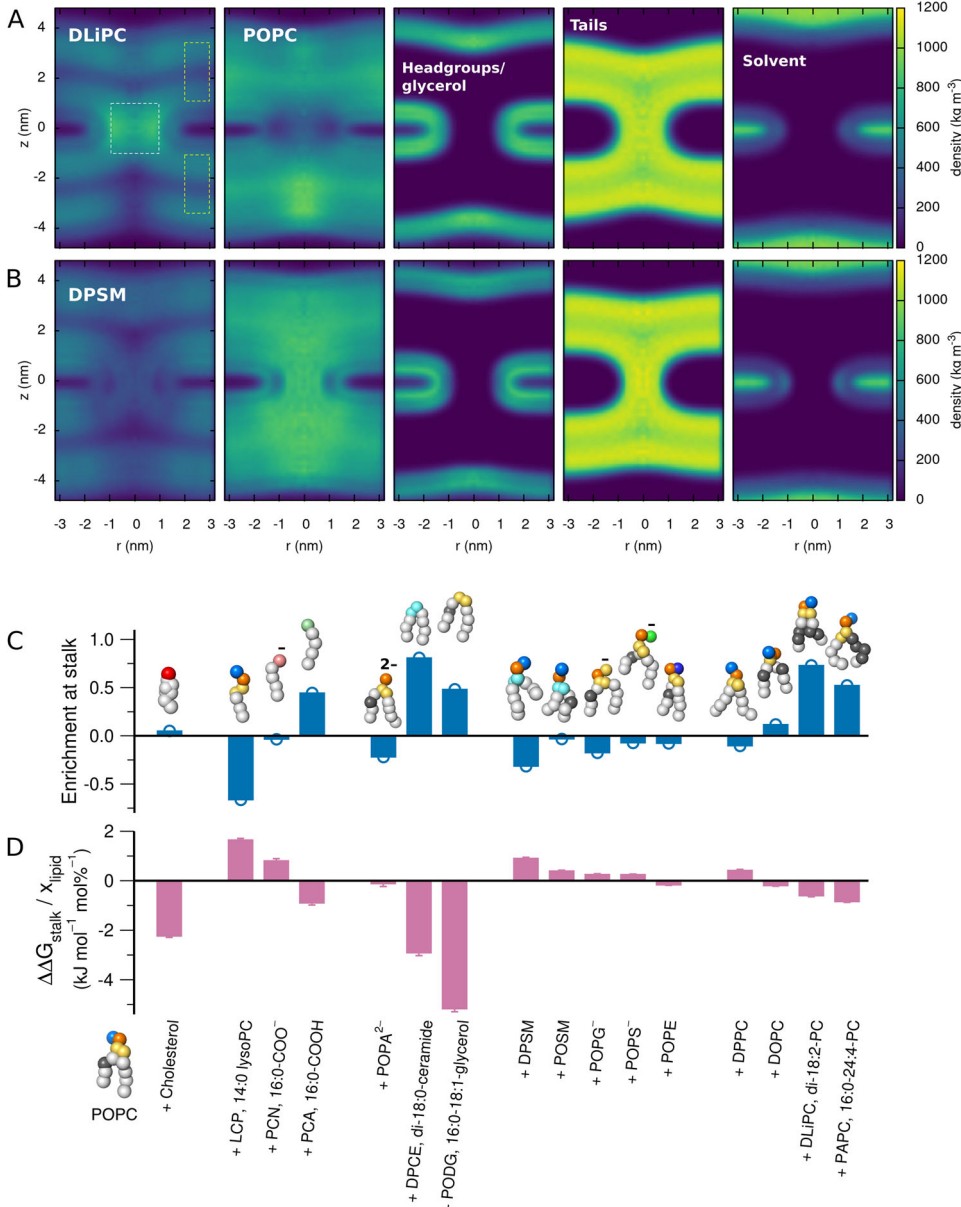

**Fig. 6 Lipid enrichment in the stalk and stalk stabilization. A** Mass densities of (from left to right) DLiPC, POPC, headgroups plus glycerol beads, lipid tails, and solvent in the fully formed stalk in a system with a POPC:DLiPC ratio of 60:40. **B** Same as in panel **A**, but for the DPSM:POPC 60:40 system. **C** Enrichment of lipids inside the stalk relative to the flat membrane: $\rho_{stalk}/\rho_{mem} - 1$, where $\rho_{stalk}$ and $\rho_{mem}$ are the average densities in the white and yellow boxes in panel **A**, respectively. Beads of Martini lipid models are colored as follows: hydrophobic saturated (white), hydrophobic unsaturated (gray), hydroxyl (red), glycerol (yellow), phosphate (orange), choline (light blue), carboxylate (pale red), carboxyl (pale green), sphingosine (cyan), serine (green), ethanolamine (dark blue). **D** Change of stalk free energy $\Delta\Delta G_{stalk}$ per lipid concentration $x_{lipid}$ upon addition of lipid to a POPC membrane, taken from the slope of a linear fit to the data in Fig. 5. Error bars (1SE) denote the uncertainty of the fitted slope. Source data are provided as a Source Data file.

weakly polar triacylglycerol, which may stabilize the stalk similar to the diacylglycerol studied here.

## Discussion

The marked difference between the outer and inner leaflet prompted us to screen free energies of stalk formation for a wide range of single-lipid membranes as well as for membranes of binary lipid mixtures. The results provided a comprehensive and quantitative fusogenicity map of lipids, as summarized in Fig. 6D. The data suggest that free energies of stalk formation are not determined by a single lipid property, but instead by a combination of molecular mechanisms: (i) Stalk formation is facilitated by increased tail disorder, as present in polyunsaturated lipids, whereas increased tail order hinders stalk formation, as common for sphingomyelin. (ii) As expected from previous studies[33], lipids with negative intrinsic curvature promote stalk formation as they may stabilize the large negative curvature of the stalk. Such mechanism applies to protonated fatty acids, to PE lipids and ceramide owing to their small head groups, as well as to polyunsaturated lipids owing to the increased tail volume. By the same token, lysoPC hinders stalk formation owing to its positive intrinsic curvature[33]. (iii) The simulations demonstrate that anionic headgroups counteract stalk formation, as implied by increased $\Delta G_{stalk}$ for PS or deprotonated fatty acids. In line with this observation, POPA hardly affects $\Delta G_{stalk}$ despite the very small headgroup. These trends are likewise

rationalized by the electrostatic repulsion between headgroups, which effectively disfavor negative curvature[42]. However, since polar and Coulombic interactions are less well represented by the Martini coarse-grained force field as compared to apolar interactions, it will be critical to test these predictions by experiments and by atomistic simulations. Complementary, it will be interesting to test the role of ionic content on $\Delta G_{stalk}$; specifically, divalent ions may bridge head groups and thereby modulate the effective curvature of the membrane.

Cholesterol strongly facilitates stalk formation. Because of its small head group and bulky polycyclic tail, cholesterol has been characterized as a cone-shaped molecule that could stabilize the negative curvature along the stalk rim[37]. Our simulations do not support this mechanism because cholesterol was only marginally enriched in the stalk. Plausible alternative mechanisms may involve the reduced hydration repulsion in the presence of cholesterol[38], as also captured by the Martini model (Supplementary Fig. 12), as well as modified chemical potentials of phospholipids in cholesterol-containing membranes. The reduced hydration repulsion facilitates the spatial approach of the two fusing membranes and, hence, likely plays an important role for the overall fusion process. For stalk formation from a given degree of hydration as studied here, however, it is more difficult to isolate the consequence of reduced hydration repulsion. On the one hand, reduced repulsion could facilitate the formation of the first contacts between the fusing leaflets, but on the other hand it may reduce the free energy of the initial lamellar state ($\xi_{ch} \approx 0.2$) and thereby render the stalk energetically less favorable. In addition to effects on hydration repulsion, cholesterol imposes increased order on the lipid tails. This may decrease the entropy of the phospholipids and, thereby, increase their chemical potentials. If so, this might facilitate the extraction of lipids from the flat membrane and into the stalk. Moreover, computed lipid densities revealed small amounts of cholesterol within the membrane core on top and below the stalk (Supplementary Fig. 13), which may stabilize the large membrane curvature, albeit this effects was greatly reduced relative to diacylglycerol (Supplementary Fig. 14). Because cholesterol exerts such diverse effects, additional studies will be required to fully resolve the mechanisms by which it facilitates stalk formation.

By simulating stalk formation between membranes with complex lipid composition, we found that the inner leaflet of a typical mammalian plasma membrane[4] is more fusogenic than the outer leaflet by ∼ 50 kJ/mol. These properties may be an adaption to evolutionary pressure: a fusogenic inner leaflet facilitates exocytosis, thereby increasing the fusion rates and reducing the required energy consumption by fusion proteins. A less fusogenic outer leaflet might hinder infection by enveloped viruses that fuse with the plasma membrane. The fusogenicity of lipids in Fig. 6D provides the molecular rationale for the distinct fusogenicities of the inner and outer leaflets. Namely, the increased fusogenicity of the inner leaflet is mainly caused by the increased number of polyunsaturated tails, mainly present as PS and PE lipids, and, to a lower degree, by the large PE content (cf. Table 1). The outer leaflet counteracts fusion mainly owing to the large sphingomyelin content, fewer polyunsaturated tails, and increased PC content relative to PE.

These conclusions were obtained with a proposed reaction coordinate for stalk formation, which allowed computationally highly efficient free energy calculations of stalk formation along thermodynamically reversible pathways. Restraints along the reaction coordinate only imposed a requested degree of connectivity between the two hydrophobic compartments, whereas the shape, radius, lateral position, and lipid composition of the stalk were controlled by the force field. Two control simulations were carried out to confirm that the pathways are reversible and, thereby, that

the free energy calculations are not affected by hysteresis problems: (i) the free energy obtained from free simulations of stalk opening and closure are in excellent agreement with the respective PMF computed with umbrella sampling (Fig. 2). (ii) Umbrella sampling simulations started from constant-velocity pulling simulations in forward and backward direction provided nearly identical PMFs (Supplementary Fig. 18). These desired properties are far from guaranteed in common PMF calculations but critically depend on the choice of a good reaction coordinate[43,44]. One PMF required less than 7 hours on an inexpensive server equipped with a 6-core CPU and a consumer graphics card, hence allowing for high-throughput calculations. Our PMF calculations complement computationally more elaborate methods such as the string method, which is capable of finding the minimum free energy path of stalk formation without the need of identifying a good reaction coordinate[27]. The computational efficiency further relied on the use of the Martini coarse-grained force field that achieves a speedup in sampling by a factor of ∼ 1000 relative to atomistic simulations due to fewer particles, longer integration time step, and accelerated lipid diffusion. However, with our reaction coordinate, free energy calculations of stalk formation with atomistic MD simulations are now within reach. This will allow us to test whether coarse-grained simulation primarily provide trends or whether they also provide quantitatively precise predictions of free energies during fusion.

While this study focused on stalk formation, it is evident that other stages of the fusion process are influenced by the lipid composition as well. Stalk widening as well as opening and expansion of the fusion pore involve highly curved membranes and are therefore controlled by lipid curvature[10,45]. Hence, future studies should quantify the role of complex lipid compositions on such later stages of fusion, in addition to lipid effects on the dehydration repulsion, which may add a free energy offset to the free energies of stalk formation computed here[27]. Further, it will be interesting to test whether lipid–protein interactions guide a local enrichment of fusogenic lipids and, thereby, further facilitate fusion.

To conclude, we presented a comprehensive and quantitative fusogenicity map of lipids. The simulations showed that the lipid composition of membranes may modulate $\Delta G_{stalk}$ by 100 kJ/mol or even more. A range of lipid properties facilitate stalk formation, including increased tail unsaturation, longer tails, smaller head groups, and cholesterol content. In contrast, stalk formation is hindered by anionic lipids and by lipids that increase the tail order such as sphingomyelin. We found that the lipid composition of the inner leaflet of a mammalian plasma membrane greatly favors stalk formation relative to the outer leaflet. The distinct fusogenicities of the two leaflets are likely an adaptation to physiological requirements and are mainly rationalized by the different content of polyunsaturated lipids, sphingomyelin, and PE lipids.

## Methods

MD simulations were carried out with Gromacs, version 2020.3[46], and with an in-house modifications of Gromacs 2018.8 and 2021.2 that implement the harmonic restraint along the reaction coordinate $\xi_{ch}$, originally introduced to study pore formation over membranes[28,29]. PMFs were computed with umbrella sampling using 19 umbrella windows and simulating each window for 200 ns. Interactions were described with the Martini coarse-grained force field version 2.2 if not stated otherwise[47]. Details on the reaction coordinate, simulation setup, parameters, and analysis are provided in the Supplementary information.

**Reporting summary**. Further information on research design is available in the Nature Research Reporting Summary linked to this article.

## Data availability
Source data used to render the figures are provided with this paper. Simulation systems (starting conformations, topologies, MD parameter files) used in this study are available in the Zenodo database at https://doi.org/10.5281/zenodo.5196036[48]. Additional data that support the findings of this study are available from the corresponding author upon reasonable request. Source data are provided with this paper.

## Code availability

The source code of the in-house modification of Gromacs that implements the chain coordinate is freely available at: https://gitlab.com/cbjh/gromacs-chain-coordinate

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

## Acknowledgements

We thank Yuliya Smirnova and Jelger Risselada for insightful discussions, for sharing the PChd200 and PCh220 simulation systems, and for comments on the manuscript. This study was supported by the Deutsche Forschungsgemeinschaft via SFB 1027/B7.

## Author contributions

C.S.P. designed and performed simulations of physiological lipid mixtures. K.C.S. performed simulations with the older POPC model. J.S.H. designed, implemented, and performed free energy calculations. C.S.P. and J.S.H. wrote the paper with contributions by K.C.S. All authors contributed to discussions of the results and approved the final version.

## Funding

## Competing interests

The authors declare no competing interests.

**Additional information**

