## [Peer Review File · Nature Communications]

REVIEWER COMMENTS

Reviewer #1 (Remarks to the Author):

The manuscript entitled “Free energies of stalk formation in the lipidomics era”, by Chetan S. Poojari, Katharina C. Scherer, and Jochen S. Hub, reports the free energy landscape of stalk formation with different lipid compositions/binary mixtures. Further, they explain how the energetics of the earliest steps of membrane fusions, a stalk formation, are controlled by membrane complexity. They have found that lipid properties such as increased tail unsaturation, longer tails, smaller head groups, and cholesterol content facilitate stalk formation. On other hand, anionic lipids and lipids that increase the tail order, such as sphingomyelin, prevent stalk formation. They also suggest that the inner leaflet of a mammalian plasma membrane has a lipid composition that favors stalk formation more than the outer leaflet.

Although the authors have computed many free energies of stalk formation in membranes with different lipid head groups, tail lengths, tail unsaturation, and sterol content to explain their findings, there are a few concerns in the methodological aspects.

1. The authors insisted that the proposed method provides a free energy profile along a reversible path way of stalk formation. In the membrane fusion, this pathway should be always clearly defined at the molecular level? In realistic process of the membrane fusion, it can most likely be non-reversible. This reviewer does not understand in which mechanism the reversibility was guaranteed in this free energy approach. This should be explained more clearly. The proposed methods only controls the local density of defined cylindrical region. How can this method guide the stalk to two separated membrane with a specific number of lipids (prepared at the beginning)? The situation should be even worse for the mixed lipids membranes. This is basically impossible. So a question is what is the reversible pathway insisted in this paper?

2. The authors have reported the PMFs of stalk formation across membranes of POPC lipids using different degrees of hydration in the proximal water compartment, as shown Fig. 1D. I can understand, the smaller intermembrane distances (less water in the proximal water compartment) make the stalk formation energetically favorable with consistent of previous reports (Ref. Biophysical Journal, 2004, 87(4) 2508–2521). However, it also shows that stalk formation has lower energy compared to the flat membranes. Hence, the authors must provide the detailed explanation regarding this. In general, stalk formation/hemi-fusion/pore formation have higher energy compare to flat membranes to prevent frequent and nonspecific spontaneous fusion events (Ref. Fig. 3 PNAS, 2017, 114 (6) 1238-1241, Biophysical Journal 2002, 82(2) 882–895). There are always specific fusion proteins needed to allow it.

3. The lower energy found at stalk rather than two separated membrane suggests a strong system-size dependency of this free energy calculations. This should be clearly commented. Also it should be clearly discussed how the authors chose the system size.

4. The radius of the cylindrical region should also affect the results. The authors assumed that the cylinder is large enough to accommodate the expected stalk structure, though it is not evident, especially when the mixed lipids are considered. In this way, the authors denied many possible other structures of stalk. This should be explained and discussed.

5. Although the authors performed the simulation with the MARTINI model 2.2/3.2, the technical results presented by the authors may not be incorrect, the authors should make sure the significance of miscibility of lipids in lipid mixtures as MARTINI lipids model fails to reproduce the phase separation experimentally observed in a DOPC/CHOL bilayer (J. Phys. Chem. B 2013, 117, 4072–4080). Such limitation can affect the obtained PMFs as reported in this work. Hence, the authors should give strong validation that current model is able to reproduce miscibility of lipids on basis of previous reports/literature or do few more analysis of different binary mixtures simulation to make sure it as supporting information. It will make more confidence of reader.

6. General comment: The authors suggested, the addition of cholesterol strongly stabilizes the stalk (page 13), and found that simulation result is qualitative agreement with experimental results (PNAS, 2012, 109, E1609–E1618). Such comparison with experimental results always make more confidence of reader. Hence, the authors should compare simulation results with experimental reports (at least qualitative) as much as possible as many experimental studies on stalk formation of membrane are available.

Reviewer #2 (Remarks to the Author):

The manuscript by Poojari et al. reports a systematic computational study on the energetics of stalk formation as a function of lipid composition. Stalk formation is the first step in vesicle fusion, crucial in a variety of biological processes. The subject is of clear interest to a broad audience in biophysics and biology.

The systematic study is enabled by a novel, fast computational technique, validated by comparison with the more sophisticated so-called "string method". Hence, the methodology used here is both solid and beyond state-of-the-art.

Many of the results and conclusions are novel, to the best of my knowledge. The claims are fully supported by the simulation results. The focus on compositional differences between inner and outer leaflet of plasma membranes is particularly interesting, as it allows making a hypothesis on the evolutionary origin of such differences. This may influence thinking in membrane biology.

Finally, the manuscript and figures are generally very clear.

Overall, I strongly support publication of this work. I have only few suggestions for possible improvements.

Minor points:

- Figure S1 is important to understand the methodology, and could be included in the main text (for example, as an extra panel in Figure 1). In Fig. S1, lipids highlighted by thicker sticks are not easily distinguished; maybe better to highlight them with a different color?

- Page 5: "With fewer than 2 water beads per lipid, a barrier or transition state emerges"; I find the sentence unclear: in the path towards fusion, a transition state always exists, but with more than 2 water beads the TS seems to be the stalk (or some other unidentified state, not visible here because only the first part of the fusion process is characterized), while with less than 2 water beads the stalk is an intermediate state (metastable) and the TS is in between the flat membrane and the stalk.

- On the comparison between free energies obtained with the string method vs. the method developed here: it not completely clear if any other simulation detail may affect the results; for example: was the lipid model the same? (An older model is shown in Figure S2).

- Link with transition state theory: how was the frequency of attempt (ν) calculated? As I understand it is simply the result of plugging k and ΔG^\ddagger (the ΔG of the transition state) into the TST (Arrhenius) equation; can it be linked to the motion of lipids in the direction of the bilayer normal? how?

- Figure 3: the hydration level is now indicated as # of water beads per nm^2 , instead of per lipid. This makes the comparison with other results (Figure 1 and 2) more problematic. However, a comparison seems necessary, to interpret the data.

- Comparison among different head groups: the results are very interesting, but I think a word of caution would be welcome when discussing them, for the following reasons: (1) the representation of charged lipids in Martini is rather crude (electrostatics is only short-ranged); (2) the representation of ions is even more crude (only hydrated ions); (3) differences among head groups are represented via (minor) differences in 1 single bead; and (4) I suspect all those beads have the same size in Martini (different from what implied at page 11), therefore it is not entirely clear how differences in size between PC and PE are captured by this force field.

- Wording: "evidently" is used a number of times in the text but it does not seem useful.

- Page 15, Effect of cholesterol: how does it affect membrane curvature? And how does it influence the dehydration repulsion? These issues should be clarified (or, references should be added).

- Page 15, effect of PODG: the similarity with triacylglycerol (TG, the main component of lipid droplets) is unclear: what would be the effect of PODG in lipid droplet formation?

Reviewed by: Luca Monticelli

Reviewer #3 (Remarks to the Author):

The authors perform Coarse-Grained (CG) Molecular Dynamics (MD) simulations to determine the free energy of fusion stalk formation between two lipid membranes. Using umbrella sampling and a weighted histogram analysis, they calculate the stalk's free energy (relative to two parallel unperturbed membranes) as a function of the hydration between the proximal leaflets (essentially, their separation). They study a wide variety of different lipid types using the MARTINI model, including some mixtures, and find that the different systems differ vastly in their readiness to fuse. Lipid membranes are more fusogenic if tails are more unsaturated, headgroups are smaller (unless they are charged), and tails are longer (at least at close distance). These findings are mostly explicable by the preference of such lipids for the stalk geometry, with a few notable exceptions (such as cholesterol). In a small but conceivably rather useful side project, they also show that the new MARTINI version 3.0beta leads to a lower stalk free energy—by the rather consistent amount of 30 kJ/mol.

These results confirm what scientists have found experimentally, but it is nice to see it being demonstrated in such an extensive and careful simulation study. I believe this work will provide valuable complementary evidence that will help to rationalize membrane fusion, and I am overall in favor of publication. I'd ask the authors to consider the following items, though:

1. In the authors' setup, hydration is essentially equivalent to distance. This tight link matters because the simulated system is so small that the membrane just a very small distance away from the stalk is essentially forced to be flat. This is different from the physiological situation. I therefore believe it would be helpful if the authors could supplement their investigation by one more scan that

probes finite size effects. Take a system that fuses well, but pick a hydration state (i.e., average distance) at which the stalk energy is higher than the initially flat membranes. Now repeat this simulation for membranes with a larger side length. Maybe twice as large, or if the authors dare even three times as large. My expectation is that this would bring the stalk energy down, since I suspect there's a sizable elastic component to it in the present setup. With a bigger membrane, a more gently inward sloping shape is possible, as opposed to a thin and long tether. Of course, I understand that the authors cannot (and in fact should not) repeat all their simulations for larger system sizes, but it would be good to illustrate to the reader what the effect of a more elastically adaptive membrane would be. If the simulations are indeed as efficient as claimed in the manuscript, this does not strike me as very expensive.

2. The authors remind us that picking a reaction coordinate is fraught with dangers, such as experiencing hysteresis, and hence it'd be much better to use a method that would not have to make any assumptions about the coordinate, such as the string method. They then proceed to pick a reaction coordinate all the same. Several of the subsequent studies help to convince me that it's a good one, but their results differ markedly from the supposedly superior string-method used in Ref. [27]. As best as I understand what the authors say (but it might help to clarify this in more than a single sentence), the problem is not in fact the reaction coordinate per se but the *end point*—the definition of the “stalk state” itself. If so, the observed discrepancy would indeed be irrelevant, since we'd be comparing apples and oranges.

3. Following on this: it would however put a bigger burden on the authors to justify their definition of “the stalk”. The way I understand it, an incredibly thin hydrophobic connecting path would suffice, its width being basically irrelevant. This makes the resulting stalk look rather different from the images one typically pictures within continuum theories. I believe it would help to expand the discussion of this point. Incidentally, the averaged density images from the SI actually have a closer appearance to those continuum models than the individual snapshots, so maybe moving one of them into the main text, or some version of it in which heads/tails/solvent are shown with different colors, might help to argue the case.

4. The authors mention several times that ionic lipids do not foster fusion, even if their head groups are small, presumably because the electrostatic repulsion makes it effectively larger. Aside from the fact that electrostatic is not handled particularly well within MARTINI, which the authors acknowledge, another point maybe worth mentioning is that this result would definitely also depend on the ionic content. Specifically, in the presence of multivalent cations (say, calcium) one could easily imagine the effect reversing, since these cations could bridge lipid head groups.

I also have a couple of minor remarks:

1. I bristle at the description of MARTINI as “near atomic”. MARTINI is firmly in the realm of coarse-grained models, since there is not a single atomic property that is not approximate by some fairly effective potential. Specifically, many properties that matter for the types of studies proposed here—hydrogen bonding in general and hydration in particular, electrostatics, tail entropy—are not actually represented all that well. I would prefer if the authors changed that descriptor in the abstract.

2. In the first paragraph on page 2 the authors write “Recent lipidomics studies showed that polyunsaturated lipids are strongly enriched in the inner leaflet.” This is needlessly vague, since the paper being cited is actually quite precise here: the cytoplasmic leaflet is about twice as unsaturated as the exoplasmic one.

3. The second paragraph on page 2 states that “Membrane fusion occurs with the help of fusion proteins because fusion requires the two opposing lipid membranes to overcome a large dehydration barrier.” While it is true that fusion proteins need to help dehydrating the fusion site, this is not the only thing they do. Since there is also no reference provided, I would recommend a bit more caution before making such a strong statement.

4. The first sentence on the third paragraph of page 2 begins with “Because intermediate structures along the fusion pathway involve highly curved membranes, [...]” It might be worthwhile to mention that *somewhere* during the fusion process the whole notion of surfaces must break down anyways.

5. In the last line on page 2 a rogue comma destroys the meaning of the sentence: “Inversely, cone-shaped unsaturated PE [...]”. The authors surely mean “Inversely cone-shaped unsaturated PE [...]”. Also, that same word, inversely, is again missing in the first line of page 3. Maybe even better, the authors could write “inverse-cone-shaped”.

6. The third line of page 3 starts with “However”, but this is confusing. What follows is not a statement that would somehow qualify what’s been said before. Rather, it supports the previous point via a second line of reasoning. I would delete it and directly write “In addition to such effects [...]”.

7. The first paragraph on page 5 repeats the cautions about a poor choice of the reaction coordinate. Since the text is almost copy-paste the same, I wonder whether it’s been accidentally left in.

8. On page 7 the authors claim that the 3 ns time between attempts corresponds approximately to the time lipids take to travel their diameter. Not only do I think that the latter time is a bit longer (maybe by one order of magnitude), but I also don't even know why that would be a particularly relevant time scale for comparison. One might rather be tempted to wonder whether lipid protrusions matter, since they initiate leaflet contact. At any rate, since it is highly dangerous to interpret time scales in CG systems, my advice would be to not hang any interpretation on those 3 ns at all. I don't think MARTINI simulations are a good basis for any such claim.

9. When the authors showed that longer tails helped to reach the stalk, I couldn't help but wonder whether this is because longer tails can more effectively "reach out" to the neighboring bilayer. Is this a legitimate picture? Would there be support in the trajectories for this? Or should readers rather be discouraged from forming such a picture in their head?

10. The authors point out that cholesterol is one of the few molecules whose support of fusogenicity doesn't follow the usual picture of partitioning into the stalk. Notably, it is (probably) also the only lipid in the mix whose flip-flop time is fast enough to change leaflets during the simulation. Is there any indication that this plays a role for the observations? Like, does the density of cholesterol across the leaflets change? If not, then it might be worthwhile to remind the reader that this is not part of the story.

11. In the last paragraph on page 13 the authors write that "[...] energetically unstable stalks are slightly thinner as compared to stalks that form the free energy minimum [...]" But since the "thickness" of a stalk is a property that's not well represented by the order parameter ξ_{ch} , I'm not sure how much weight I'm allowed to put on this statement.

12. In the second-to-last paragraph in the Results section on page 15 the authors write "These findings suggest that cholesterol does not favor stalk formation owing to an intrinsic negative curvature of cholesterol, but more likely owing to its influence on the dehydration repulsion between the membranes." I'm not sure on what basis the authors make their assessment of "more likely" (other than I guess that anything is more likely than something that definitely doesn't happen). But they make a rather specific guess here, and I wonder why they talk about that one, rather than other ones.

13. In line 8 of page 16, the authors write "A counter example is given by lysoPC [...]" This is potentially confusing, because they do not produce a counterexample to the basic principle they have been talking about. That lysoPC, as a positively curved lipid, would not support stalk formation is expected within the authors' model. I'd rather write "By the same token, lysoPC hinders stalk formation [...]", or "On the flip side, lysoPC hinders stalk formation [...]"

14. The reaction coordinate ξ_{ch} is fairly carefully defined in the SI. What's missing is a definition of the smeared indicator function $f(r)$. For completeness, it'd be nice if the authors include that information as well.

Dear Reviewers,

Many thanks for your constructive and highly insightful comments, which helped us to strongly improve our manuscript.

Please find our point-by-point response below. Your comments are colored in black and our replies in blue. We further provide PDFs of the main text and SI that highlight the modifications in the text.

In response to an editorial request, we have taken several measures to optimize the re-use of our data:

- 1) We published the in-house modified Gromacs, which implements the chain coordinate, together with a test case for stalk formation, on GitLab at:

<https://gitlab.com/cbjh/gromacs-chain-coordinate>

<https://gitlab.com/cbjh/gromacs-chain-coordinate/-/tree/main/documentation-chain-coord/Example-stalk-formation>

- 2) We uploaded our simulation systems (structures, topologies, MD parameter files) to zenodo.org (doi: 10.5281/zenodo.5196035). During the review process, the archive can be accessed anonymously via the following secret link:

<https://zenodo.org/record/5196036?token=eyJhbGciOiJIUzUxMiIsImV4cCI6MTYzMTQ4Mzk5OSwiaWF0IjoxNj40ODUyNzQxfQ.eyJkYXRhIjp7InJlY2lkIjo1MTk2MDM2fSwiaWQiOiE2NDkxLkYybWQiOiIyZjUxNjhkZCJ9.oUMAvxb6Lv3AUihXeSi7Va1wzlmHSblrSFIL0gZeUg1UGzZqJVeMo8v5OXK3difPBX3LxUxpaxc>

The archive will become Open Access if the manuscript is accepted.

- 3) We added a Supporting Information zipped folder with the source data of the manuscript figures.

Sincerely yours,

Chetan Poojari
Katharina Scherer
Jochen Hub

Reviewer #1 (Remarks to the Author):

The manuscript entitled "Free energies of stalk formation in the lipidomics era", by Chetan S. Poojari, Katharina C. Scherer, and Jochen S. Hub, reports the free energy landscape of stalk formation with different lipid compositions/binary mixtures. Further, they explain how the energetics of the earliest steps of membrane fusions, a stalk formation, are controlled by membrane complexity. They have found that lipid properties such as increased tail unsaturation, longer tails, smaller head groups, and cholesterol content facilitate stalk formation. On other hand, anionic lipids and lipids that increase the tail order, such as sphingomyelin, prevent stalk formation. They also suggest that the inner leaflet of a mammalian plasma membrane has a lipid composition that favors stalk formation more than the outer leaflet.

Although the authors have computed many free energies of stalk formation in membranes with different lipid head groups, tail lengths, tail unsaturation, and sterol content to explain their findings, there are a few concerns in the methodological aspects.

REVIEWER:

1. The authors insisted that the proposed method provides a free energy profile along a reversible path way of stalk formation. In the membrane fusion, this pathway should be always clearly defined at the molecular level? In realistic process of the membrane fusion, it can most likely be non-reversible. This reviewer does not

understand in which mechanism the reversibility was guaranteed in this free energy approach. This should be explained more clearly. [...]

REPLY: We fully agree that fusion in a biological context is typically non-reversible. In order to understand the energetics of the process, it is nevertheless insightful to study the associated free energy landscape, which is by definition an equilibrium property. With "reversible" in our PMF calculations, we mean that the system takes the *same pathway* for stalk opening and stalk closing as we pull along our reaction coordinate (RC) in forward and backward direction. This property is critical to avoid hysteresis problems during PMF calculations and, thereby, to obtain well-defined free energy differences. We demonstrate this desired property with two test simulations: (i) We carried out pulling simulations in both directions (stalk opening and closing) and started umbrella sampling simulations from the respective frames. We find excellent agreement between the two PMFs (Fig. S18), as expected for reversible pathways. (ii) Free simulations of stalk opening and closing are in excellent agreement with the PMFs (Fig. 2), showing that the free energies are correct and well-defined.

This favorable behavior is by far not guaranteed in common pulling simulations but critically depends on the choice of a *good* reaction coordinate (which is often far from trivial). Indeed, we originally designed our RC because we struggled with massive hysteresis problems in simulations of pore formation in membranes when using previously proposed reaction coordinates (Awasthi and Hub, JCTC 2016, <http://dx.doi.org/10.1021/acs.jctc.6b00369>). Notably, the test presented above for the correctness of the free energies and the absence of hysteresis (hence, the reversibility) are often not done (or not shown) in other MD simulations studies, probably because these tests would reveal that the applied reaction coordinate is not very good.

To make these points clearer, we now write in the Discussion:

"These conclusions were obtained with a novel reaction coordinate for stalk formation, which allowed computationally highly efficient free energy calculations of stalk formation along thermodynamically reversible pathways. Two control simulations were carried out to confirm that the pathways are reversible and, thereby, that the free energy calculations are not affected by hysteresis problems: (i) the free energy obtained from free simulations of stalk opening and closure are in excellent with the respective PMF computed with umbrella sampling (Fig. 2). (ii) Umbrella sampling simulations started from constant-velocity pulling simulations in forward and backward direction provided nearly identical PMFs (Fig. S18). These desired properties are by far not guaranteed in common PMF calculations but critically depend on the choice of a good reaction coordinate [43,44]."

REVIEWER:

[...] The proposed methods only controls the local density of defined cylindrical region. How can this method guide the stalk to two separated membrane with a specific number of lipids (prepared at the beginning)? The situation should be even worse for the mixed lipids membranes. This is basically impossible. So a question is what is the reversible pathway insisted in this paper?

REPLY: Our RC controls (or steers) only the degree of connectivity between the two hydrophobic compartments. How the connectivity is obtained is fully controlled by the force field. For instance, we do not impose the shape, radius, or lateral position of the stalk, or which lipid type is used to form the stalk. Instead, the force field decides which stalk structure and composition is most favorable (has lowest free energy) at a given degree of connectivity. We now write:

"Restrains along the reaction coordinate only imposed a requested degree of connectivity between the two hydrophobic compartments, whereas the shape, radius, lateral position, and lipid composition of stalk were controlled by the force field."

In this study, we do not discuss the details of the pathway of stalk formation, such as the mechanism of "lipid spaying", which has been discussed by others in detail (e.g., the work by Smirnova or Risselada).

REVIEWER:

2. The authors have reported the PMFs of stalk formation across membranes of POPC lipids using different degrees of hydration in the proximal water compartment, as shown Fig. 1D. I can understand, the smaller intermembrane distances (less water in the proximal water compartment) make the stalk formation energetically favorable with consistent of previous reports (Ref. Biophysical Journal, 2004, 87(4) 2508–2521). However, it also shows that stalk formation has lower energy compared to the flat membranes. Hence, the authors must provide the detailed explanation regarding this. In general, stalk formation/hemi-fusion/pore formation have higher energy compare to flat membranes to prevent frequent and nonspecific spontaneous fusion events (Ref. Fig. 3 PNAS, 2017, 114 (6) 1238-1241, Biophysical Journal 2002, 82(2) 882–895). There are always specific fusion proteins needed to allow it.

REPLY: Thank you for pointing this out. At very low hydration, the flat membranes are already in a state of high free energy owing to the hydration repulsion (which is reproduced by the MARTINI models in a semi-quantitative manner, see also new Fig. S12). Hence, upon forming the stalk, part of the dehydration repulsion is relieved, thereby making stalk formation increasingly favorable. To clarify this point, we now write:

"Such negative $\Delta G[\text{stalk}]$ values at very low hydration are rationalized by the energy stored in the hydration repulsion between the two membranes. Upon stalk formation, the hydration repulsion energy is partly relieved, thereby decreasing the free energy of the stalk relative to the flat membranes."

In addition, we have added the reference pointed out the reviewer. Thank you for pointing this out.

REVIEWER:

3. The lower energy found at stalk rather than two separated membrane suggests a strong system-size dependency of this free energy calculations. This should be clearly commented. Also it should be clearly discussed how the authors chose the system size.

REPLY: We are particular thankful for this point, which was also raised by Reviewer 3. Previously, we tested the role of system size only for pure phospholipid system (not shown in the previous manuscript). In response to this comment, we carried out a more extensive scan that also included cholesterol-containing membranes, for which finite size effects could be more pronounced (owing to their increased rigidity). The new Figure S1 shows PMFs of stalk formation for three different lipid compositions (DOPC, DIPC, DOPC:Cholesterol 70:30) and the respective stalk free energies $\Delta G[\text{stalk}]$. Evidently, $\Delta G[\text{stalk}]$ for pure-phospholipid membranes is hardly influenced by finite-size effects (Fig. S1D, black, yellow). With cholesterol, in contrast, very small systems lead to increased $\Delta G[\text{stalk}]$, suggesting that slightly larger systems (compared to pure-phospholipid membranes) are required to avoid finite-size artifacts on $\Delta G[\text{stalk}]$.

These results prompted to recompute all PMFs for cholesterol-containing membranes (including the physiological mixtures shown in Fig. 3), now using a fixed box dimension in X- and Y-direction of 7.5nm rather than a fixed number of 64 lipids per monolayer as used previously. In consequence, $\Delta G[\text{stalk}]$ values for cholesterol-containing membranes decreased considerably relative to the previous manuscript. The modified numbers do not change any of the key findings of this study, but role of cholesterol as an important determinant for fusion is further emphasized.

As an additional test, we further recomputed the PMFs for systems with single-chained lipids (lyso-PC and oleic acid) likewise with a fixed box area instead of a fixed number of lipids. Because these $\Delta G[\text{stalk}]$ values hardly changed, we kept the previous setup.

REVIEWER:

4. The radius of the cylindrical region should also affect the results. The authors assumed that the cylinder is large enough to accommodate the expected stalk structure, though it is not evident, especially when the mixed lipids are considered. In this way, the authors denied many possible other structures of stalk. This should be explained and discussed.

REPLY: This is a misunderstanding. The radius of the cylinder does not impose a certain radius of the stalk. The radius of the stalk is purely decided by the force field. To avoid this misunderstanding, we now add Figure S17 (showing PMFs computed with different cylinder radii) and we write explicitly in the SI:

"Critically, the radius of the cylinder of 1.2 nm does **not** control the radius of the stalk. Instead, the cylinder is merely used to ensure the locality of the hydrophobic protrusions in the membrane plane. If the cylinder radius would be too large, two laterally displaced hydrophobic protrusions, one from the upper and one from the lower membrane, could be misinterpreted as a continuous hydrophobic connection. Such mechanism would further allow the membrane to evade the energetically unfavorable transition state of stalk formation, which could lead to undesired hysteresis effects. Figure S17 shows that the PMFs depend only marginally on the choice of the cylinder radius in the range between 1.2nm and 1.6nm."

REVIEWER:

5. Although the authors performed the simulation with the MARTINI model 2.2/3.2, the technical results presented by the authors may not be incorrect, the authors should make sure the significance of miscibility of lipids in lipid mixtures as MARTINI lipids model fails to reproduce the phase separation experimentally observed in a DOPC/CHOL bilayer (J. Phys. Chem. B 2013, 117, 4072–4080). Such limitation can affect the obtained PMFs as reported in this work. Hence, the authors should give strong validation that current model is able to reproduce miscibility of lipids on basis of previous reports/literature or do few more analysis of different binary mixtures simulation to make sure it as supporting information. It will make more confidence of reader.

REPLY: Phase separation does (to our knowledge) only occur in ternary mixtures, such as DOPC/sphingomyelin/cholesterol or DOPC/DPPC/cholesterol, as studied in the reference pointed out by the reviewer (J. Phys. Chem. B 2013, 117, 4072–4080). Binary mixtures as studied here should not reveal phase separation. Physiological mixtures should likewise not reveal phase separation (the existence of "rafts" in the plasma membrane remains highly controversial). In response to this remark, we visually inspected several simulation systems and did not detect any phase separation, in line with the expectation from experiments. We now write in the SI:

"Visual inspection of the simulation, also after long simulation times, did not reveal any indication of phase separation or lipid demixing (see e.g. Fig. S5)."

REVIEWER:

6. General comment: The authors suggested, the addition of cholesterol strongly stabilizes the stalk (page 13), and found that simulation result is qualitative agreement with experimental results (PNAS, 2012, 109, E1609–E1618). Such comparison with experimental results always make more confidence of reader. Hence, the authors should compare simulation results with experimental reports (at least qualitative) as much as possible as many experimental studies on stalk formation of membrane are available.

REPLY: Thank you for pointing this out we have added several references to original experimental studies, specifically on the role of PE head groups and diacylglycerol.

Reviewer #2 (Remarks to the Author):

The manuscript by Poojari et al. reports a systematic computational study on the energetics of stalk formation as a function of lipid composition. Stalk formation is the first step in vesicle fusion, crucial in a variety of biological processes. The subject is of clear interest to a broad audience in biophysics and biology.

The systematic study is enabled by a novel, fast computational technique, validated by comparison with the more sophisticated so-called "string method". Hence, the methodology used here is both solid and beyond state-of-the-art.

Many of the results and conclusions are novel, to the best of my knowledge. The claims are fully supported by the simulation results. The focus on compositional differences between inner and outer leaflet of plasma

membranes is particularly interesting, as it allows making a hypothesis on the evolutionary origin of such differences. This may influence thinking in membrane biology. Finally, the manuscript and figures are generally very clear.

Overall, I strongly support publication of this work. I have only few suggestions for possible improvements.

Minor points:

- Figure S1 is important to understand the methodology, and could be included in the main text (for example, as an extra panel in Figure 1). In Fig. S1, lipids highlighted by thicker sticks are not easily distinguished; maybe better to highlight them with a different color?

REPLY: We agree that the illustration of the "cylinder" would be useful in the main text. We have therefore now added the cylinder illustration (together with the labels for the proximal water compartment and the proximal leaflet) to Figure 1 and removed the previous Figure S1 to avoid redundancy.

REVIEWER:

- Page 5: "With fewer than 2 water beads per lipid, a barrier or transition state emerges"; I find the sentence unclear: in the path towards fusion, a transition state always exists, but with more than 2 water beads the TS seems to be the stalk (or some other unidentified state, not visible here because only the first part of the fusion process is characterized), while with less than 2 water beads the stalk is an intermediate state (metastable) and the TS is in between the flat membrane and the stalk.

REPLY: We agree that the sentence was imprecise. We now avoid the term "emerge" and write more precisely:

"With fewer than 2 water beads per lipid, the stalk is separated from the flat membrane by a barrier or transition state, indicating that the stalk is metastable (long-living)."

REVIEWER:

- On the comparison between free energies obtained with the string method vs. the method developed here: it not completely clear if any other simulation detail may affect the results; for example: was the lipid model the same? (An older model is shown in Figure S2).

REPLY: Since the difference relative to the free energy profile from the string method puzzled us as well, we used exactly the same simulation system used by Smirnova et al., which was kindly provided by the authors of this study (same force field, same older POPC model shown in Fig S2C, and MD parameter (mdp) options).

The difference most likely comes from the different definition of the "stalk state" (see also our response to Reviewer 3). With our method, the stalk is defined with $\xi_{ch} \sim 1$, which merely imposed a continuous hydrophobic connection between the two membranes, but this connection may take any radius, shape, or lateral position. Smirnova et al. defined the stalk with a specific 3D density of the hydrophobic beads, which allows much less conformational freedom. We believe that their "stalk definition" does not include other possible realizations of a stalk (e.g., at different lateral positions), thereby overestimating the free energy of the stalk. We have clarified these points in the Figure caption of Fig. S2.

REVIEWER:

- Link with transition state theory: how was the frequency of attempt (ν) calculated? As I understand it is simply the result of plugging k and ΔG^\ddagger (the DG of the transition state) into the TST (Arrhenius) equation; can it be linked to the motion of lipids in the direction of the bilayer normal? how?

REPLY: Yes, the attempt frequency ν was simply obtained via $k(\text{TST}) = \nu * \exp(-\Delta G^\ddagger/kT)$, using ΔG^\ddagger from the PMF and the rates from the free, unbiased simulations. We now write more clearly:

"The rates from the free simulations together with the barrier from the PMF imply $\nu = \dots$ "

In addition, also in response to Reviewer 3, we have analyzed the motions of the head groups along the z-direction, which likewise resembles the observed time scale of 3ns (new Fig. S19 and new SI Discussion). In the main text, we now write:

"Interestingly, this time scale resembles to the time scales required for (i) head groups to rearrange along the membrane normal and (ii) for lipids to travel a typical lipid-lipid distance (Fig. S19 and SI Discussion). Hence, the attempts for stalk formation in the context of transition state theory may be interpreted as rearrangements owing to conformational sampling of lipids, which occur on the nanosecond timescale."

REVIEWER:

- Figure 3: the hydration level is now indicated as # of water beads per nm², instead of per lipid. This makes the comparison with other results (Figure 1 and 2) more problematic. However, a comparison seems necessary, to interpret the data.

REPLY: Thank you for pointing this out. We now write in the legends of Figs 1 and 2:

"With increasing hydration in the proximal compartment of 1.25 to 7 water beads per lipid, corresponding to 4 to 21.2 water beads per nm²" and "...and 1.8 water beads per lipid between the proximal leaflets (5.6 water beads per nm²)"

REVIEWER:

- Comparison among different head groups: the results are very interesting, but I think a word of caution would be welcome when discussing them, for the following reasons: (1) the representation of charged lipids in Martini is rather crude (electrostatics is only short-ranged); (2) the representation of ions is even more crude (only hydrated ions); (3) differences among head groups are represented via (minor) differences in 1 single bead; and (4) I suspect all those beads have the same size in Martini (different from what implied at page 11), therefore it is not entirely clear how differences in size between PC and PE are captured by this force field.

REPLY: We fully agree that electrostatic interactions are quite crude in Martini, hence any effects on stalk formation based on electrostatics must be taken with care. We state these limitations now more clearly (replaced "will be interesting" with "will be critical"):

"However, since polar and Coulombic interactions are less well represented by the Martini coarse-grained force field as compared to apolar interactions, it will be critical to test these predictions by experiments and by atomistic simulations."

Regarding the head group sizes: The head groups (PC, PE, PS, and PG differ indeed only by one bead type (Q0, Qd, P5, P4, respectively), whose Lennard-Jones potentials all have the same sigma parameter (0.47nm). Nevertheless, when comparing DOPC with DOPE, there is a clear preference for DOPE for negative curvature (we see this is several projects). The effective size of the head groups (and, thereby, the effective shape of the lipid) is tuned only with the epsilon parameter of the bead type at the tip of head group, as listed here for the interests of the reviewer:

Lipid	atomtypes	sigma	eps
DOPC	Q0-Q0	0.47	3.5
DOPE	Qd-Qd	0.47	5
POPS	P5-P5	0.47	5.6
POPG	P4-P4	0.47	5

Hence, the larger epsilon of DOPE relative to DOPC (5 vs. 3.5 kJ/mol) makes the DOPE headgroup effectively smaller, possibly by reducing the magnitude of thermal fluctuations of Qd-Qd distances relative to Q0-Q0 distances.

For PG and PS relative to PC, the trend is less clear because the different charge changes the head group size (in addition to a different epsilon-parameter). Therefore, we now added a table to the SI, that lists the areas per lipid of DOPC, DOPE, DOPS, and DOPG, as the area per lipid may be taken as a qualitative measure for head group size (at fixed tail types). We also now write more carefully:

"These findings cannot be explained merely the geometric shape of the lipids because PS, PG, and PC headgroups have a similar size in the Martini model, as evident from very similar areas per lipid (Tab. S2)."

REVIEWER:

- Wording: "evidently" is used a number of times in the text but it does not seem useful.

REPLY: Thank you for pointing this out, as we fully agree. We removed "evidently" at several occasions or replaced with a more specific expression, such as "Figure 6C/D reveals that, ..."

REVIEWER:

- Page 15, Effect of cholesterol: how does it affect membrane curvature? And how does it influence the dehydration repulsion? These issues should be clarified (or, references should be added).

REPLY: In response to this comment, we have largely rewritten both sections on cholesterol (in Results and Discussion). As additional data, we now computed a set of PMFs for dehydrating the proximal leaflet to quantify the effect of cholesterol on hydration repulsion (new Fig. S12, Methods described in the SI Methods). Because cholesterol takes effect by possibly multiple mechanisms, we added a more extensive discussion to page 17 of the main text and a statement, that "additional studies will be required to fully resolve the mechanisms by which it [cholesterol] facilitates stalk formation. "

REVIEWER:

- Page 15, effect of PODG: the similarity with triacylglycerol (TG, the main component of lipid droplets) is unclear: what would be the effect of PODG in lipid droplet formation?

REPLY: The similarity is that di- and triacylglycerol are both only weakly polar, suggesting that triacylglycerol may stabilize stalks in a similar manner as diacylglycerol/PODG studied here. We now write:

"We hypothesize that a similar mechanism applies during biogenesis of lipid droplets, which are released from the ER membrane via a stalk-like intermediate.[36] Lipids droplets contain large amounts of only weakly polar triacylglycerol, which may stabilize the stalk similar to the diacylglycerol studied here"

Reviewed by: Luca Monticelli

Reviewer #3 (Remarks to the Author):

The authors perform Coarse-Grained (CG) Molecular Dynamics (MD) simulations to determine the free energy of fusion stalk formation between two lipid membranes. Using umbrella sampling and a weighted histogram analysis, they calculate the stalk's free energy (relative to two parallel unperturbed membranes) as a function of the hydration between the proximal leaflets (essentially, their separation). They study a wide variety of different lipid types using the MARTINI model, including some mixtures, and find that the different systems differ vastly in their readiness to fuse. Lipid membranes are more fusogenic if tails are more unsaturated, headgroups are smaller (unless they are charged), and tails are longer (at least at close distance). These findings are mostly explicable by the preference of such lipids for the stalk geometry, with a few notable exceptions (such as cholesterol). In a small but conceivably rather useful side project, they also show that the new MARTINI version 3.0beta leads to a lower stalk free energy—by the rather consistent amount of 30 kJ/mol.

These results confirm what scientists have found experimentally, but it is nice to see it being demonstrated in such an extensive and careful simulation study. I believe this work will provide valuable complementary evidence that will help to rationalize membrane fusion, and I am overall in favor of publication. I'd ask the authors to consider the following items, though:

1. In the authors' setup, hydration is essentially equivalent to distance. This tight link matters because the simulated system is so small that the membrane just a very small distance away from the stalk is essentially forced to be flat. This is different from the physiological situation. I therefore believe it would be helpful if the authors could supplement their investigation by one more scan that probes finite size effects. Take a system that fuses well, but pick a hydration state (i.e., average distance) at which the stalk energy is higher than the initially flat membranes. Now repeat this simulation for membranes with a larger side length. Maybe twice as large, or if the authors dare even three times as large. My expectation is that this would bring the stalk energy down, since I suspect there's a sizable elastic component to it in the present setup. With a bigger membrane, a more gently inward sloping shape is possible, as opposed to a thin and long tether. Of course, I understand that the authors cannot (and in fact should not) repeat all their simulations for larger system sizes, but it would be good to illustrate to the reader what the effect of a more elastically adaptive membrane would be. If the simulations are indeed as efficient as claimed in the manuscript, this does not strike me as very expensive.

REPLY: Thank you for this important suggestion, which was also raised by Reviewer 1. We have now computed PMFs of stalk formation for three lipid compositions (pure DOPC, pure DIPC, and DOPC:cholesterol 70:30) at various membrane sizes (new Figure S1). We find that the membranes without cholesterol are remarkably robust against finite size effects, as the $\Delta G[\text{stalk}]$ hardly changes with increasing box size. For cholesterol-containing membranes, however, smaller boxes render the stalk more unstable, likely because cholesterol makes the membranes more rigid, leading to increased effects owing to the elastic component pointed out by the reviewer.

Based on this new analysis, we decided to recompute all PMFs for cholesterol-containing membranes (including the physiological mixtures) with larger boxes, defined by a box length of 7.5nm in X- and Y-directions. All trends and conclusions that we drew from the PMFs before are maintained, but the $\Delta G[\text{stalk}]$ values decreased considerably in presence of cholesterol. We updated all panels of Fig. 3, Fig. 5A, 6C/D, and S5. We further added a section "Number of lipids per leaflet and membrane size" to the Methods.

REVIEWER:

2. The authors remind us that picking a reaction coordinate is fraught with dangers, such as experiencing hysteresis, and hence it'd be much better to use a method that would not have to make any assumptions about the coordinate, such as the string method. They then proceed to pick a reaction coordinate all the same. Several of the subsequent studies help to convince me that it's a good one, but their results differ markedly from the supposedly superior string-method used in Ref. [27]. As best as I understand what the authors say (but it might help to clarify this in more than a single sentence), the problem is not in fact the reaction coordinate per se but the *end point*—the definition of the "stalk state" itself. If so, the observed discrepancy would indeed be irrelevant, since we'd be comparing apples and oranges.

REPLY: Yes, this is 100% correct, as the problem is the definition of the end points. Our "stalk state" is defined by $\xi_{\text{ch}} = 1$, which enforces "some" hydrophobic connection between the two membranes, but the stalk radius, lateral position, or lipid composition are freely sampled by the force field. In the method by Smirnova *et al.* (PNAS 2019), the stalk is defined by a specific 3D density distribution of the hydrophobic beads, which excluded all other possible realizations of the stalk (e.g., at a different lateral position). In addition, the authors reported that they typically suppress fluctuations by restraining the 3D density (Ref. 2 of the SI). In consequence, the method by Smirnova *et al.* lead to a higher free energy of the stalk. Since we validated our $\Delta G[\text{stalk}]$ with free simulations, we must conclude that our $\Delta G[\text{stalk}]$ provides the correct probability for finding a stalk relative to finding a flat membrane.

As such, the beauty of the string method is to find the minimum free energy path *between* two previously defined end states. But if the end states are different, one obviously arrives at different free energy differences.

Nevertheless, the study by Smirnova et al. is methodologically excellent and pushes the limit of what can be studied with the (quite demanding) string method. Therefore, there is no reason to criticize this study. Instead, we now write more explicitly:

"The difference of ~ 30 kJ/mol may be rationalized by the different definitions of the end states. In our method, the stalk end state ($\xi_{\text{ch}} \approx 1$) is defined by the presence of a hydrophobic connection between the two membranes, but all possible shapes, radii, and lateral positions of the connection are included in the state with $\xi_{\text{ch}} \approx 1$. In Ref. 1, the stalk state is defined with a specific 3D density of the hydrophobic beads, which may allow fewer conformational states than the stalk definition adopted by us."

REVIEWER:

3. Following on this: it would however put a bigger burden on the authors to justify their definition of "the stalk". The way I understand it, an incredibly thin hydrophobic connecting path would suffice, its width being basically irrelevant. This makes the resulting stalk look rather different from the images one typically pictures within continuum theories. I believe it would help to expand the discussion of this point. Incidentally, the averaged density images from the SI actually have a closer appearance to those continuum models than the individual snapshots, so maybe moving one of them into the main text, or some version of it in which heads/tails/solvent are shown with different colors, might help to argue the case.

REPLY: Yes, we define the stalk as a state with a continuous hydrophobic connection between the membranes, but the radius of the stalk is fully controlled by the force field. In consequence, the radius of the final stalk differs considerably among different simulations - very fusogenic systems prefer thicker stalks, as one may expect (see, e.g., the stalk with di-18:3-PC in Fig. S8, or some of the stalks for the inner plasma membrane leaflet in the updated Figure S5). The very thin stalks referred to by the reviewer are rather a feature of unstable stalks, which would anyway rapidly close in a free simulation.

The stalks in our simulations may not be identical to stalks modelled with continuum models, possibly because the lipid conformations seem more disordered inside the stalk compared to the cartoon figures in earlier continuum studies (see, e.g., Figs 1 and 2 in Kozlovsky and Kozlov, BJ 2002, [https://doi.org/10.1016/S0006-3495\(02\)75450-7](https://doi.org/10.1016/S0006-3495(02)75450-7)). Such increased tail disorder may allow for thinner stalks. However, the overall shape of the stalk is in reasonable agreement with previous simulation studies of metastable stalks (see e.g., Katsov, Mueller, Schick, BJ 2004, <https://doi.org/10.1529/biophysj.103.038943>). Therefore, we trust that our definition for having a stalk ($\xi_{\text{ch}} \approx 1$) does not impose any unphysical stalk structure.

To clarify these points, we now write:

"The overall shape of the stalk is largely independent of the lipid composition, except that energetically unstable stalks are thinner as compared to stalks that form the free energy minimum. The shapes reasonably agree with previous simulation studies [18], but the stalks obtained here --even if metastable-- are slightly thinner as compared to stalks modeled by continuum descriptions [15], possibly because the lipids were highly disordered inside the stalk in our simulations."

REVIEWER:

4. The authors mention several times that ionic lipids do not foster fusion, even if their head groups are small, presumably because the electrostatic repulsion makes it effectively larger. Aside from the fact that electrostatic is not handled particularly well within MARTINI, which the authors acknowledge, another point maybe worth mentioning is that this result would definitely also depend on the ionic content. Specifically, in the presence of multivalent cations (say, calcium) one could easily imagine the effect reversing, since these cations could bridge lipid head groups.

REPLY: Thank you for this point. Since the weakness of MARTINI with respect to Coulombic interactions was stated but possibly not emphasized enough, we now write more clearly:

However, since polar and Coulombic interactions are less well represented by the Martini coarse-grained force field as compared to apolar interactions, it will be **critical** [instead of "interesting"] to test these predictions by experiments and by atomistic simulations."

The role of ionic content is outside the scope of this study, but an interesting point for the discussion and exciting for future studies. We added:

"Complementary, it will be interesting to test the role of ionic content on ΔG [stalk]; specifically, divalent ions may bridge head groups and thereby modulate the effective curvature of the membrane."

REVIEWER:

I also have a couple of minor remarks:

1. I bristle at the description of MARTINI as "near atomic". MARTINI is firmly in the realm of coarse-grained models, since there is not a single atomic property that is not approximate by some fairly effective potential. Specifically, many properties that matter for the types of studies proposed here—hydrogen bonding in general and hydration in particular, electrostatics, tail entropy—are not actually represented all that well. I would prefer if the authors changed that descriptor in the abstract.

REPLY: We agree that the notion "near atomic" could be mistaken as a euphemism. We replaced the notion with "coarse-grained".

REVIEWER:

2. In the first paragraph on page 2 the authors write "Recent lipidomics studies showed that polyunsaturated lipids are strongly enriched in the inner leaflet." This is needlessly vague, since the paper being cited is actually quite precise here: the cytoplasmic leaflet is about twice as unsaturated as the exoplasmic one.

REPLY: Thank you for pointing this out. We now write: "Recent lipidomics studies showed that polyunsaturated lipids are approximately twice as abundant in the inner leaflet compared to the outer leaflet."

REVIEWER:

3. The second paragraph on page 2 states that "Membrane fusion occurs with the help of fusion proteins because fusion requires the two opposing lipid membranes to overcome a large dehydration barrier." While it is true that fusion proteins need to help dehydrating the fusion site, this is not the only thing they do. Since there is also no reference provided, I would recommend a bit more caution before making such a strong statement.

REPLY: We agree that this statement was imprecise. Since we do not want to discuss the roles of fusion proteins, which is not the topic of this study, we now write more generally:

"Fusion involves high kinetic barriers because it requires to overcome the hydration repulsion between the membranes and to form intermediates with highly curved membranes. Cells and viruses use fusion proteins to overcome these kinetic barriers [6,8]."

REVIEWER:

4. The first sentence on the third paragraph of page 2 begins with "Because intermediate structures along the fusion pathway involve highly curved membranes, [...]" It might be worthwhile to mention that *somewhere* during the fusion process the whole notion of surfaces must break down anyways.

REPLY: Since we do not consider pore formation here (where the surface picture clearly breaks down), we are not sure whether stressing the picture of a "surface" is helpful for the readers. Surfaces are more inherent to continuum models, but we have purely particle simulations. Hence, we prefer at this point to stress the freedom of the author and not mention "surface".

REVIEWER:

5. In the last line on page 2 a rogue comma destroys the meaning of the sentence: "Inversely, cone-shaped unsaturated PE [...]". The authors surely mean "Inversely cone-shaped unsaturated PE [...]". Also, that same word, inversely, is again missing in the first line of page 3. Maybe even better, the authors could write "inverse-cone-shaped".

REPLY: This is a misunderstanding. PE lipids are cone-shaped, not "inverted cone-shaped", due to their small head group and bulky tails. However, we agree that "Inversely" can trigger this misunderstanding. We now write instead: "In contrast, cone-shaped PE lipids...".

REVIEWER:

6. The third line of page 3 starts with "However", but this is confusing. What follows is not a statement that would somehow qualify what's been said before. Rather, it supports the previous point via a second line of reasoning. I would delete it and directly write "In addition to such effects [...]".

REPLY: We fully agree, thank you for spotting this. Fixed.

REVIEWER:

7. The first paragraph on page 5 repeats the cautions about a poor choice of the reaction coordinate. Since the text is almost copy-paste the same, I wonder whether it's been accidentally left in.

REPLY: We agree. We now shortened the beginning of the paragraph and only write:

"To avoid hysteresis problems during potential of mean force (PMF) calculations, it is mandatory to define a good reaction coordinate."

REVIEWER:

8. On page 7 the authors claim that the 3 ns time between attempts corresponds approximately to the time lipids take to travel their diameter. Not only do I think that the latter time is a bit longer (maybe by one order of magnitude), but I also don't even know why that would be a particularly relevant time scale for comparison. One might rather be tempted to wonder whether lipid protrusions matter, since they initiate leaflet contact. At any rate, since it is highly dangerous to interpret time scales in CG systems, my advice would be to not hang any interpretation on those 3 ns at all. I don't think MARTINI simulations are a good basis for any such claim.

REPLY: Thank you for this comment. To clarify these points, also in response to a comment by Reviewer 2, we added additional analysis and discussion to the SI (See "Supporting Information Discussion" and Fig. S19). These new data and discussion clarify several important aspects raised by the reviewer:

1) It is indeed known that rates/dynamics are too fast with MARTINI because coarse-graining leads to a smoothed energy landscape. The lateral diffusion is typically accelerated by a factor of 4, but other dynamics may be accelerated by other factors. As such, the value of 3ns, which we obtained via transition state theory (TST), should be taken as the time scale **within the MARTINI model**, which is likely decreased compared to atomistic simulations. However, since the value of 3ns is anyway only an estimate (an error of 1kT in the barrier would change the computed time scale by a factor of $e=2.718$), the increased dynamics is only a moderate additional correction. We now write:

"It is important to note that, because the Martini model leads to a smoothed energy landscape, all dynamics discussed here (for lateral and normal displacements as well as for stalk formation) are likely accelerated relative to atomistic simulations or to experimental conditions. [with ref. to Marrink, Tieleman, Chem Soc Rev, 2013]"

2) In response to this comment, we now computed the lateral diffusion coefficient for a pure POPC MARTINI membrane, taken from the slope of the mean-square displacement. We obtained $D = 0.073 \text{ nm}^2/\text{s}$. The

approximate lipid-lipid distance is $\Delta r = \sqrt{A_L}$, where $A_L \approx 0.58 \text{ nm}^2$ is the area per lipid of the Martini POPC model. Using $\langle \Delta r^2 \rangle = 4D \cdot \Delta t$ (two-dimensional diffusion), we obtain as a typical time for travelling a lipid-lipid distance as

$$\Delta t = \langle \Delta r^2 \rangle / 4D \approx 2 \text{ ns.}$$

Hence, the time scales are indeed similar (within the Martini model). For atomistic simulations, this time scale is expected to be $\sim 4x$ larger.

3) As suggested by the reviewer, we now analyzed the time scale for lipid motions along the membrane normal (z direction), which may be interpreted as attempts for "lipid protrusions" to trigger stalk formation (new Fig. S19). Not too surprisingly, the normal rearrangements occur on a similar time scale (of 2-3ns) as the lateral lipid displacements. This merely reflects that lipids rearrange their lipid-lipid contacts on this time scale, which leads to both lateral and normal rearrangements. With the new analysis, we now show explicitly that the normal lipid displacements may indeed be interpreted as "attempts" (in the context of TST) for stalk formulation, as anticipated by the Reviewer. Hence, the new analysis provides a much more direct link between (i) the lipid conformational dynamics and (ii) the attempt frequency from TST.

In addition to the new SI Discussion and new Fig. S19, we now write in the main text:

"Interestingly, this time scale resembles to the time scales required for (i) head groups to rearrange along the membrane normal and (ii) for lipids to travel a typical lipid-lipid distance (Fig. S19 and SI Discussion). Hence, the attempts for stalk formation in the context of transition state theory may be interpreted as rearrangements owing to conformational sampling of lipids, which occurs on the nanosecond timescale."

REVIEWER:

9. When the authors showed that longer tails helped to reach the stalk, I couldn't help but wonder whether this is because longer tails can more effectively "reach out" to the neighboring bilayer. Is this a legitimate picture? Would there be support in the trajectories for this? Or should readers rather be discouraged from forming such a picture in their head?

REPLY: We also thought extensively about such "simple" geometric pictures. However, such picture would not explain why the $\Delta G[\text{stalk}]$ increases with tail length at large hydration, whereas $\Delta G[\text{stalk}]$ decreases with tail length for low hydration (see Fig. S7 and 4B). Hence, other effects likely play a role, such as different hydration repulsion, or different intrinsic curvature. Very long bulky tails at large hydration could also hinder the lipids from shielding the hydrophobic core of the stalk from the proximal water. As the effects of longer tails are probably complex (while only moderately relevant for biological systems), we prefer to simply report the effect and not to speculate on the mechanism in this case.

REVIEWER:

10. The authors point out that cholesterol is one of the few molecules whose support of fusogenicity doesn't follow the usual picture of partitioning into the stalk. Notably, it is (probably) also the only lipid in the mix whose flip-flop time is fast enough to change leaflets during the simulation. Is there any indication that this plays a role for the observations? Like, does the density of cholesterol across the leaflets change? If not, then it might be worthwhile to remind the reader that this is not part of the story.

REPLY: We agree that cholesterol flip-flop deserves additional attention. For instance, cholesterol could be enriched in the inner leaflet and thereby further facilitate stalk formation. We have now carried out two additional types of analysis:

1) We analyzed the flip-flop events between the proximal and distal leaflets during the 200ns stalk-opening simulations of the four POPC/cholesterol systems. We find typically 5 to 10 flip-flop events during 200ns for each flip-flop direction (~ 20 to ~ 40 in total). The direction of flip-flops seem entirely random, that is, there is no preference for distal-to-proximal vs. proximal-to-distal flip-flop. Hence, it seems unlikely that cholesterol is enriched in the inner leaflet and, thereby, facilitates stalk formation.

2) If cholesterol enrichment in one leaflet would play a role, we would expect that the PMFs gradually change with longer simulation time of the umbrella windows. However, by looking at PMFs from time blocks of 50ns (50-100ns, 100-150ns, etc., which we do anyway in all our projects to test for convergence), there is no trend visible. Instead, the PMFs from time blocks agree within few kJ/mol. Hence, it seems unlikely that slow flip-flop affects the PMFs in a systematic way. We now write in the SI:

"In addition to the bootstrapping analysis, we computed PMFs from 50ns time blocks of the umbrella simulations (50-100ns, 100-150ns, 150-200ns). The PMFs agreed within few kilojoules per mole, giving additional support to the convergence of the PMFs. In addition, this analysis suggests that flip-flop events of cholesterol, which occurs on long time scales, could systematically bias the PMFs of cholesterol-containing systems."

REVIEWER:

11. In the last paragraph on page 13 the authors write that "[...] energetically unstable stalks are slightly thinner as compared to stalks that form the free energy minimum [...]" But since the "thickness" of a stalk is a property that's not well represented by the order parameter ξ_{ch} , I'm not sure how much weight I'm allowed to put on this statement.

REPLY: Please note our comments above on the thickness of the stalks. In brief, the reaction coordinate ξ_{ch} does not control the radius of the stalk. The radius of the stalk is fully controlled by the force field (at a given degree of "connectivity" between the membranes), hence the simulation adopts the thickness that has the lowest free energy (at a given degree of connectivity).

REVIEWER:

12. In the second-to-last paragraph in the Results section on page 15 the authors write "These findings suggest that cholesterol does not favor stalk formation owing to an intrinsic negative curvature of cholesterol, but more likely owing to its influence on the dehydration repulsion between the membranes." I'm not sure on what basis the authors make their assessment of "more likely" (other than I guess that anything is more likely than something that definitely doesn't happen). But they make a rather specific guess here, and I wonder why they talk about that one, rather than other ones.

REPLY: We agree that this statement requires a clearer justification, as also suggested by Reviewer 2.

To quantify the effect of cholesterol on hydration repulsion, we have now computed PMFs of dehydrating the proximal water compartment between membranes with 0%, 20% and 40% cholesterol content (new Fig. S12, new SI paragraph "PMF calculations of dehydration of the proximal leaflet"). We find that cholesterol indeed strongly decreases the hydration repulsion within Martini, in qualitative agreement with experiments. Clearly, such result should be considered as "trend" because hydration repulsion is influenced by electrostatic interactions and entropy, which are both not particularly well represented by Martini.

Further, we have largely rewritten the two paragraphs on cholesterol (in Results and Discussion), please see our response to Reviewer 2 above.

REVIEWER:

13. In line 8 of page 16, the authors write "A counter example is given by lysoPC [...]" This is potentially confusing, because they do not produce a counterexample to the basic principle they have been talking about. That lysoPC, as a positively curved lipid, would not support stalk formation is expected within the authors' model. I'd rather write "By the same token, lysoPC hinders stalk formation [...]", or "On the flip side, lysoPC hinders stalk formation [...]"

REPLY: Thank you for spotting this. We're happy to adopt the Reviewer's suggestion "By the same token".

REVIEWER:

14. The reaction coordinate x_{i_ch} is fairly carefully defined in the SI. What's missing is a definition of the smeared indicator function $f(r)$. For completeness, it'd be nice if the authors include that information as well.

REPLY:

We agree. We have added a section "Definition of the differentiable indicator function" in the SI Methods.

REVIEWERS' COMMENTS

Reviewer #1 (Remarks to the Author):

The authors have addressed my concerns satisfactorily.

Reviewer #2 (Remarks to the Author):

In the revised version of the manuscript, the authors addressed all criticism by the reviewers. Text and figures are now clearer, and new analyses are included in Supporting Info.

Overall, I support publication of the manuscript as is.

Reviewer #3 (Remarks to the Author):

The authors have very carefully addressed the comments of all three reviewers and have revised their manuscript accordingly. It has become clearer and more compelling in the process, and I have no remaining scientific concerns about its publication. I specifically thank the authors for including the study shown in Supplementary Figure 1. This is convincing evidence that the system sizes chosen are by and large big enough (which slightly surprised me!). I am happy for the authors that while doing this additional work they caught the problem that cholesterol-containing systems require somewhat larger boxes, and I commend them for undertaking the significant effort of re-running all these simulations with a larger lipid number.

I congratulate the authors to this well-done study, which I trust will meet a receptive audience. At this point, I have only one minor wording issue and a bunch of small language nit-picks left, which I point out for the authors' convenience.

Page 2: "polyunsaturated lipids are approximately twice as abundant in the inner leaflet compared to the outer leaflet." The authors wrote this in response to my request to be more precise about the degree of unsaturation. Unfortunately, this is now too specific and not quite right, since it suggests that the NUMBER of unsaturated lipids in the cytoplasmic leaflet is double that in the outer leaflet.

What Lorent et al. rather wrote is the following: “the abundance-weighted average unsaturation is approximately twofold greater for cytoplasmic leaflet phospholipids”. I think a good concise compromise would be to write that the average degree of unsaturation in the cytoplasmic leaflet is about twice that of the outer leaflet.

Page 7, bottom: “Interestingly, this time scale resembles to the time scales required for [...]” The “to” is too much.

Page 17: “Plausible alternative mechanisms may involve the reduced hydration repulsion in presence of cholesterol [...]” Make it “[...] in THE presence of [...]”

Page 18, top: “It is conceivable that such effect facilitates the extraction of lipids from the flat membrane and into the stalk.” Maybe better: “It is conceivable that this facilitates [...]” Or alternatively, “If so, this might facilitate [...]”

Page 18: “Because cholesterol takes such diverse effects, [...]” Maybe replace “takes” by “exerts” or “exhibits”.

Page 19, top: “the free energy obtained from free simulations of stalk opening and closure are in excellent AGREEMENT with the respective PMF [...]”

Page 19: “These desired properties are by far not guaranteed [...]”. Might be better as “These desired properties are far from guaranteed [...]”

Dear Reviewers,

Thank you again for this exceptionally constructive review process, which was indeed a pleasure for us. Thank you for the encouraging feedback.

Please find below our reply to the language suggestions by Reviewer 3. We also attached two PDFs diff.pdf and diff_SI.pdf, which highlight the latest modifications in the manuscript.

Sincerely,

Chetan Poojari
Katharina Scherer
Jochen Hub

Reviewer #1 (Remarks to the Author):

The authors have addressed my concerns satisfactorily.

Reviewer #2 (Remarks to the Author):

In the revised version of the manuscript, the authors addressed all criticism by the reviewers. Text and figures are now clearer, and new analyses are included in Supporting Info. Overall, I support publication of the manuscript as is.

Reviewer #3 (Remarks to the Author):

The authors have very carefully addressed the comments of all three reviewers and have revised their manuscript accordingly. It has become clearer and more compelling in the process, and I have no remaining scientific concerns about its publication. I specifically thank the authors for including the study shown in Supplementary Figure 1. This is convincing evidence that the system sizes chosen are by and large big enough (which slightly surprised me!). I am happy for the authors that while doing this additional work they caught the problem that cholesterol-containing systems require somewhat larger boxes, and I commend them for undertaking the significant effort of re-running all these simulations with a larger lipid number.

I congratulate the authors to this well-done study, which I trust will meet a receptive audience. At this point, I have only one minor wording issue and a bunch of small language nit-picks left, which I point out for the authors' convenience.

Page 2: "polyunsaturated lipids are approximately twice as abundant in the inner leaflet compared to the outer leaflet." The authors wrote this in response to my request to be more precise about the degree of unsaturation. Unfortunately, this is now too specific and not quite right, since it suggests that the NUMBER of unsaturated lipids in the cytoplasmic leaflet is double that in the outer leaflet. What Lorent et al. rather wrote is the following: "the abundance-weighted average unsaturation is approximately twofold greater for cytoplasmic leaflet phospholipids". I think a good concise compromise would be to write that the average degree of unsaturation in the cytoplasmic leaflet is about twice that of the outer leaflet.

Page 7, bottom: "Interestingly, this time scale resembles to the time scales required for [...]" The "to" is too much.

Page 17: "Plausible alternative mechanisms may involve the reduced hydration repulsion in presence of cholesterol [...]" Make it "[...] in THE presence of [...]"

Page 18, top: "It is conceivable that such effect facilitates the extraction of lipids from the flat membrane and into the stalk." Maybe better: "It is conceivable that this facilitates [...]" Or alternatively, "If so, this might facilitate [...]"

Page 18: “Because cholesterol takes such diverse effects, [...]” Maybe replace “takes” by “exerts” or “exhibits”.

Page 19, top: “the free energy obtained from free simulations of stalk opening and closure are in excellent AGREEMENT with the respective PMF [...]”

Page 19: “These desired properties are by far not guaranteed [...]”. Might be better as “These desired properties are far from guaranteed [...]”

REPLY: Thank you for spotting these issues. We have adopted all the reviewer’s suggestions.